# Jaw shape and mechanical advantage are indicative of diet in Mesozoic mammals

Nuria Melisa Morales-García [1✉], Pamela G. Gill[1,2], Christine M. Janis [1,3] & Emily J. Rayfield [1]

Jaw morphology is closely linked to both diet and biomechanical performance, and jaws are one of the most common Mesozoic mammal fossil elements. Knowledge of the dietary and functional diversity of early mammals informs on the ecological structure of palaeo-communities throughout the longest era of mammalian evolution: the Mesozoic. Here, we analyse how jaw shape and mechanical advantage of the masseter (MAM) and temporalis (MAT) muscles relate to diet in 70 extant and 45 extinct mammals spanning the Late Triassic-Late Cretaceous. In extant mammals, jaw shape discriminates well between dietary groups: insectivores have long jaws, carnivores intermediate to short jaws, and herbivores have short jaws. Insectivores have low MAM and MAT, carnivores have low MAM and high MAT, and herbivores have high MAM and MAT. These traits are also informative of diet among Mesozoic mammals (based on previous independent determinations of diet) and set the basis for future ecomorphological studies.

[1] School of Earth Sciences, Wills Memorial Building, University of Bristol, Bristol, UK. [2] Department of Earth Sciences, Natural History Museum, London, UK. [3] Department of Ecology and Evolutionary Biology, Brown University, Providence, RI, USA. ✉email: nm15309@my.bristol.ac.uk

Our understanding of Mesozoic mammals has dramatically improved in the past three decades. Once thought to have been conservative in locomotory modes and dietary preferences, Mesozoic mammals are now considered to have been more ecologically diverse[1–3]. In a similar fashion, it was thought that they were restricted to small sizes (<5 kg), but some taxa, like *Repenomamus giganticus* (approx. 12–14 kg), indicate greater body size diversity among Mesozoic mammals[1,2,4]. Fossils of complete skeletons reveal a diversity of locomotor lifestyles, including swimming, gliding, fossorial, and arboreal forms; craniodental morphology also indicates a diversity of diets (see refs. [5,6] and references therein). The majority of the evolutionary history of mammals (approx. 127 million years [65%]) took place during the Mesozoic[1], and so the study of Mesozoic mammal evolution also underpins our understanding of their later radiation. Although the most abundant remains of Mesozoic mammals are their teeth, lower jaws are also relatively common. Study of jaw shape and jaw biomechanics can increase our understanding of their dietary and functional evolution, and has the potential to contribute to our knowledge of the ecological structure of Mesozoic mammalian communities, in turn aiding our understanding of the prevailing vegetation and climatic conditions[7].

Ecomorphological analyses, which study potentially predictive relationships between organismal morphology and ecology (see ref. [8] for review), are one approach to the study of the dietary preferences of Mesozoic mammals. Such analyses have widely been used in mammals; in particular, the correlation of jaw morphology with dietary preferences. For example, the position of the condyle with respect to the tooth row[9], the dimensions of the jaw (e.g., the length of the diastema and the coronoid process, and the depth of jaw ramus)[10], and the predominance of one or other of the adductor muscles[11] have all been used to inform on diet. With respect to Mesozoic mammals: morphometry-driven approaches include landmark-based geometric morphometrics studies on jaw shape (e.g., see ref. [12]), and functionally-informed studies include analyses of jaw ratios (e.g., see ref. [13]), skull and jaw mechanics and tooth wear[14–16]. For example, Grossnickle and Polly[12] compared the jaw shapes of extant and Mesozoic mammals and found a clear separation between the more herbivorous multituberculates and other, more faunivorous Mesozoic taxa. Gill et al.[14] employed a suite of biomechanical techniques to show diverging dietary preferences in an Early Jurassic faunal assemblage, between stem mammals *Morganucodon* and *Kuehneotherium*. More recently, Grossnickle[17] analyzed a comprehensive set of functional metrics in the jaws of extant therian mammals and identified a set of characteristics distinguishing herbivorous from faunivorous taxa across different clades, including the size of the angular process and the length of the posterior portion of the jaw.

Here, we use a combination of morphometric-driven and functionally-driven approaches to study how mechanical advantage (used as a proxy for adductor muscle performance) and jaw shape relate to diet in Mesozoic mammals and small extant mammals. Mechanical advantage is the ratio of the length of the in-lever (i.e., moment arm of the muscle) divided by the length of the out-lever (i.e., distance from the jaw condyle to the biting point)[14,18], and so is a measure of the performance of the adductor muscles (i.e., how much force is produced at the bite point as a result of force being input by the muscles). A high mechanical advantage indicates a jaw optimized for bite force, while a low mechanical advantage indicates a jaw optimized for closure at speed. This metric has been used to study adductor muscle performance in Cenozoic mammals (including extant taxa) such as carnivorans (e.g., see ref. [19]), rodents (e.g., see ref. [20]), and bats (e.g., see ref. [21]), among others. Comparative mechanical advantage of the jaw (or a similar biomechanical

metric) has been used as a proxy for prey choice and feeding ecology in stem mammals[14], to study the yaw and pitch of the jaws of Mesozoic therian mammals and relatives[22], and to analyze ecomorphological disparity during the Mesozoic/Cenozoic transition[13].

The aim of our study is to determine whether jaw shape and mechanical advantage of the jaws of small mammals can be used as an ecomorphological proxies to elucidate the dietary preferences and behavior of Mesozoic taxa. While the jaw shapes of many multituberculates indicate a herbivorous or omnivorous diet[12], there is no clear consensus on the diets of many Mesozoic taxa typically considered as "generalized insectivores". Here we include only such generalized taxa (Fig. 1) and exclude multituberculates and haramiyidans (i.e., allotherians)[23]. We use extant taxa of small mammals of known diets to explore whether jaw shape and mechanical advantage can be suitable proxies for diet in Mesozoic mammals. A list of taxa used in this study is presented in Table 1.

## Results

**Jaw shape variation and diet in small mammals.** Using 2D geometric morphometrics (Fig. 2a), we found that jaw shape is a good proxy for diet among small extant mammals. In Fig. 3, taxa with negative PC1 scores have shorter jaws, and taxa with positive PC1 scores have longer jaws; taxa with positive PC2 scores have taller ascending rami and taxa with negative PC2 scores have shorter ascending rami. Among extant mammals, most dietary categories (excluding omnivores) can be distinguished along PC1 (Fig. 3a): herbivores plot at the negative end of PC1, insectivores towards the positive end, and carnivores in between. These categories are also statistically different from each other (Table 2), showing that jaw shape can distinguish between most major dietary types. However, our data cannot distinguish between carnivores and omnivores.

Data on the jaw shape of Mesozoic mammals were projected onto the extant taxa morphospace (Fig. 3b). In order to determine whether jaw shape could be used as a dietary proxy in Mesozoic mammals, we obtained previous independent determinations of likely diets, which variously employed dental morphology, tooth wear facets and body size (e.g., see refs. [1,7,12,14,24–32]). We saw a very good correspondence between previous proposed diets for Mesozoic mammals and their position on the morphospace. See Supplementary Fig. 6 for a principal components analysis scatter plot which includes multituberculates and haramiyidans; these taxa were excluded from our study because the vast majority of them have jaw shapes dissimilar to the other extinct and extant mammals in our sample (i.e., allotherians have shorter jaws and thus more negative PC1 scores).

*Stem mammals.* Most stem mammals plot within the morphospace of extant insectivores and have positive PC1 scores. One exception is *Sinoconodon* (taxon #2, Fig. 3), which plots within the morphospace of extant carnivores; *Sinoconodon* is considered a carnivore based on dental morphology[5]. *Haramiyavia* (#1) is thought to have been a plant-dominated omnivore[23] based on dental morphology, but here it plots within the morphospace of extant insectivores. Both morganucodontans in this study, *Morganucodon* (#3) and *Dinnetherium* (#4), have similar PC1 scores to extant insectivores, echoing the findings of Gill et al.[14].

Molar morphology indicates omnivorous or faunivorous diets for docodontans; here they mostly plot within the morphospace of extant insectivores, with the exception of *Haldanodon* (#6) and *Docofossor* (#7). *Agilodocodon* (#9) was previously considered a plant-dominated omnivore, with exudativorous dental features which indicated a diet mainly composed of plant sap[33]; more

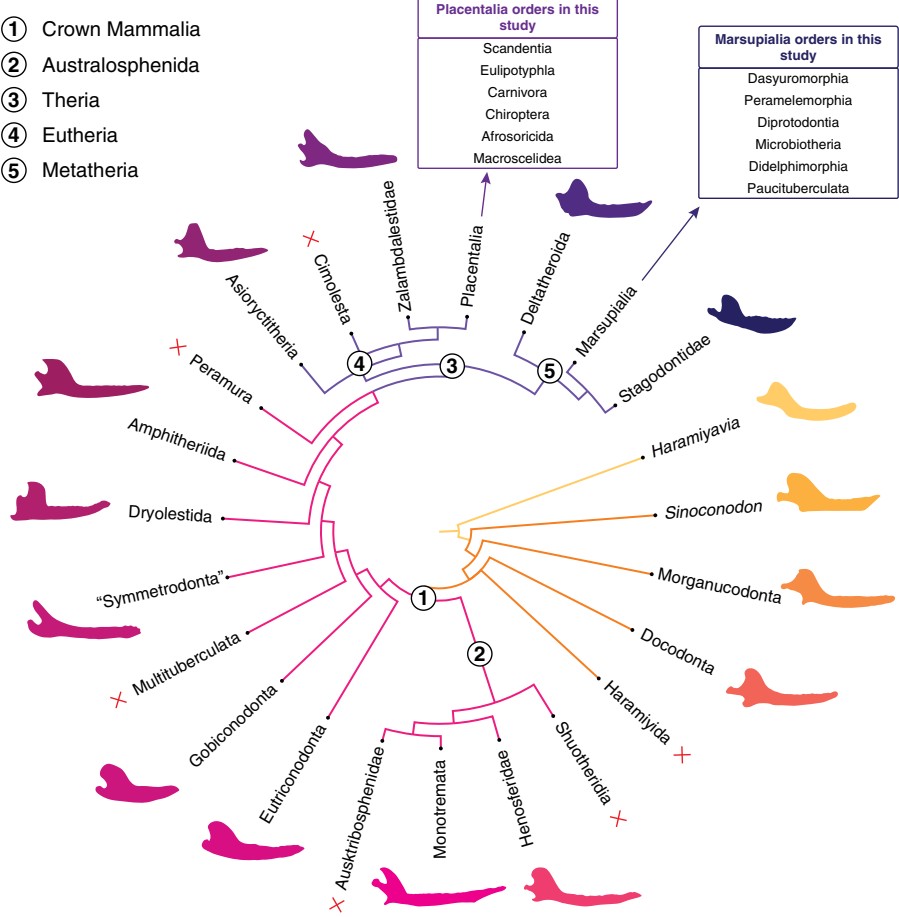

**Fig. 1 Summary of the phylogeny used in this study.** Overall topology from refs. [60],[61]. Other references in Methods section. Red crosses indicate clades not included in the study.

recently, Wible and Burrows[34] challenged this hypothesis and suggested that the teeth of *Agilodocodon* most closely resemble those of extant insectivores. Here, *Agilodocodon* plots firmly within the morphospace of extant insectivores, close to the insectivorous dusky antechinus (*Antechinus swainsonii*, #61) and the elephant shrews (*Elephantulus rufescens* [#114] and *E. brachyrhynchus* [#115]), which are insect-dominated omnivores.

According to Ji et al.[28] the swimming docodontan, *Castorocauda* (#5), has dental features indicative of feeding on aquatic invertebrates and small vertebrates, like fish. *Castorocauda* is often depicted as being carnivorous and, particularly, piscivorous[7,28,33]. The jaw shape of *Castorocauda* is similar to that of modern day insectivores; this docodontan might have been feeding on "soft" aquatic invertebrates (Fig. 3). The other Mesozoic mammal in our sample proposed to have been semi-aquatic, *Teinolophos* (#13), plots in a similar area of the morphospace to *Castorocauda*. Our extant sample also includes a semi-aquatic carnivore, the water opossum (*Chironectes minimus*, #69), which plots in the middle of the carnivore morphospace, far away from *Castorocauda* and *Teinolophos*.

*Docofossor* (#7) has skeletal features indicative of a fossorial lifestyle and a dentition similar to those of extant mammals foraging underground, such as moles, solenodons, and tenrecs[35]. This docodontan has previously been considered an insectivore[7]. Here, *Docofossor* plots within the morphospace of extant carnivores; however, it plots close to the burrowing Hispaniolan solenodon (*Solenodon paradoxus*, #109), which has an insectivorous diet. Among the extant insectivores in our sample, the burrowing vermivores (e.g., the hairy-tailed mole, *Parascalops*

*breweri* [#108], and the Hispaniolan solenodon) have more negative PC1 scores than other insectivores (similar to that of *Docofossor*), and their PC1 values are more similar to those of carnivores.

The dental morphology of *Haldanodon* (#6) is indicative of an insectivorous diet. Here, it plots within the carnivore morphospace (very near extant herbivores), because of its tall coronoid process and comparatively shorter jaw. *Docodon* (#8) likely ate insects and other small invertebrates[27] and, based on its diminutive size[36], *Microdocodon* (#10) was probably insectivorous. Both of these docodontans plot within the insectivore morphospace.

*Non-therian crown mammals.* The jaw shape of non-therian crown mammals varies widely, plotting mostly within the morphospace of insectivores and carnivores. *Fruitafossor* (#11), a fossorial mammal with teeth similar to extant armadillos, has been considered an omnivore eating insects, small invertebrates and some plants[26]. Here, it plots within the insectivore morphospace, closely to the insectivorous and fossorial hairy-tailed mole (*Parascalops breweri*, #108), and shares similar PC1 scores with other fossorial taxa, such as *Docofossor* (#7) and the Hispaniolan solenodon (#109).

Extant monotremes eat insects and other small invertebrates. It has been proposed that the Early Cretaceous monotreme *Teinolophos* (#13) had a semiaquatic lifestyle (on the basis of its enlarged mandibular canal[37]) and ate in a similar manner to the insectivorous *Kuehneotherium*[38]. Here *Teinolophos*, and the australosphenidan *Henosferus* (#12), have PC1 scores similar to insectivores and omnivores.

**Table 1 Complete list of all the taxa used in this study.**

| Extinct mammals | | Extant mammals | | | |
|---|---|---|---|---|---|
| **Stem mammals** | | **Marsupialia** | | **Placentalia** | |
| 1 | *Haramiyavia* | **Diprotodontia** | | **Scandentia** | |
| 2 | *Sinoconodon* | 46 | *Bettongia penicillata* | 75 | *Tupaia splendidula* |
| 3 | *Morganucodon* | 47 | *Potorous tridactylus* | 76 | *Tupaia dorsalis* |
| 4 | *Dinnetherium* | 48 | *Thylogale billardierii* | 77 | *Tupaia glis* |
| 5 | *Castorocauda* | 49 | *Dendrolagus goodfellowi* | **Carnivora** | |
| 6 | *Haldanodon* | 50 | *Dorcopsulus vanheurni* | 78 | *Nandinia binotata* |
| 7 | *Docofossor* | 51 | *Petaurus breviceps* | 79 | *Leopardus wiedii* |
| 8 | *Docodon* | 52 | *Pseudocheirus peregrinus* | 80 | *Lynx rufus* |
| 9 | *Agilodocodon* | 53 | *Acrobates pygmaeus* | 81 | *Felis margarita* |
| 10 | *Microdocodon* | 54 | *Phalanger orientalis* | 82 | *Prionodon pardicolor* |
| **Non-therian crown mammals** | | 55 | *Trichosurus vulpecula* | 83 | *Genetta genetta* |
| 11 | *Fruitafossor* | **Dasyuromorphia** | | 84 | *Paradoxurus hermaphroditus* |
| 12 | *Henosferus* | 56 | *Dasyurus hallucatus* | 85 | *Fossa fossana* |
| 13 | *Teinolophos* | 57 | *Dasyurus geoffroii* | 86 | *Galidia elegans* |
| 14 | *Phascolotherium* | 58 | *Sarcophilus harrisi* | 87 | *Crossarchus oscurus* |
| 15 | *Yanoconodon* | 59 | *Parantechinus apicalis* | 88 | *Herpestes javanicus* |
| 16 | *Triconodon* | 60 | *Phascogale tapoatafa* | 89 | *Lycalopex griseus* |
| 17 | *Trioracodon* | 61 | *Antechinus swainsonii* | 90 | *Vulpes corsac* |
| 18 | *Volaticotherium* | 62 | *Antechinus flavipes* | 91 | *Mephitis macroura* |
| 19 | *Argentoconodon* | 63 | *Sminthopsis crassicaudata* | 92 | *Conepatus humboldtii* |
| 20 | *Gobiconodon* | 64 | *Planigale ingrami* | 93 | *Nasua narica* |
| 21 | *Repenomamus* | 65 | *Myrmecobius fasciatus* | 94 | *Bassaricyon gabbii* |
| 22 | *Spalacotherium* | **Peramelemorphia** | | 95 | *Procyon lotor* |
| 23 | *Origolestes* | 66 | *Perameles bougainville* | 96 | *Taxidea taxus* |
| 24 | *Zhangheotherium* | **Microbiotheria** | | 97 | *Eira barbara* |
| 25 | *Maotherium* | 67 | *Dromiciops gliroides* | 98 | *Lontra canadensis* |
| 26 | *Crusafontia* | **Didelphimorphia** | | 99 | *Mustela nivalis* |
| 27 | *Amblotherium* | 68 | *Philander andersoni* | **Chiroptera** | |
| 28 | *Amphitherium* | 69 | *Chironectes minimus* | 100 | *Pteropus vampyrus* |
| 29 | *Vincelestes* | 70 | *Metachirus nudicaudatus* | 101 | *Noctilio leporinus* |
| **Therian crown-mammals** | | 71 | *Marmosa robinsoni* | 102 | *Artibeus jamaiciensis* |
| 30 | *Deltatheridium* | 72 | *Monodelphis americana* | 103 | *Chrotopterus auritus* |
| 31 | *Didelphodon* | 73 | *Caluromys derbianus* | 104 | *Myotis lucifugus* |
| 32 | *Eodelphis* | **Paucituberculata** | | 105 | *Plecotus auritus* |
| 33 | *Alphadon* | 74 | *Rhyncholestes raphanurus* | **Eulipotyphla** | |
| 34 | *Sinodelphys* | | | 106 | *Blarina brevicauda* |
| 35 | *Juramaia* | | | 107 | *Atelerix albiventris* |
| 36 | *Eomaia* | | | 108 | *Parascalops breweri* |
| 37 | *Maelestes* | | | 109 | *Solenodon paradoxus* |
| 38 | *Asioryctes* | | | **Afrosoricida** | |
| 39 | *Sasayamamylos* | | | 110 | *Microgale cowani* |
| 40 | *Kennalestes* | | | 111 | *Microgale brevicaudata* |
| 41 | *Daulestes* | | | 112 | *Tenrec ecaudatus* |
| 42 | *Uchkudukodon* | | | 113 | *Potamogale velox* |
| 43 | *Kulbeckia* | | | **Macroscelidea** | |
| 44 | *Barunlestes* | | | 114 | *Elephantulus rufescens* |
| 45 | *Zalambdalestes* | | | 115 | *Elephantulus brachyrhynchus* |

Taxa numbers used in Figs. 3, 6, and 7. Bold letters indicate the clade or group taxa belong to.

The eutriconodontans are a very diverse group of insectivores and carnivores which had a wide range of body sizes, including some of the largest Mesozoic mammals known[1]. Here all eutriconodontans fall within or very close to the extant carnivore morphospace. In particular, *Triconodon* (#16) and *Argentoconodon* (#19) plot within the carnivore morphospace, *Trioracodon* (#17) and *Volaticotherium* (#18) plot between the carnivore and insectivore morphospaces, and *Yanoconodon* (#15) plots within the insectivore morphospace. Both gobiconodontids, *Gobiconodon* (#20) and *Repenomamus* (#21), have more negative PC1 scores and plot closer to the herbivore morphospace, but still remain within or close to the carnivore morphospace. *Triconodon*, *Trioracodon*, *Gobiconodon*, and *Repenomamus* are all considered carnivores based on craniodental morphology and body size[1,7,31]; additionally, there is direct evidence for the carnivorous diet of *Repenomamus* from fossilized stomach contents[4]. *Yanoconodon* and *Volaticotherium* are considered insectivores[7].

"Symmetrodontans" like *Spalacotherium* (#22), *Zhangheotherium* (#24) and *Maotherium* (#25) have often been considered insectivores based on their craniodental morphology[1,7] (note "symmetrodontans" likely do not represent a monophyletic group, but are often grouped together based on their tooth morphology[1]). Here, all "symmetrodontans" plot within the insectivore morphospace. Dryolestids are also commonly considered insectivorous[1,29]. Here, *Crusafontia* (#26) plots between the morphospace of extant carnivores and insectivores, while *Amblotherium* (#27) plots within the insectivore morphospace.

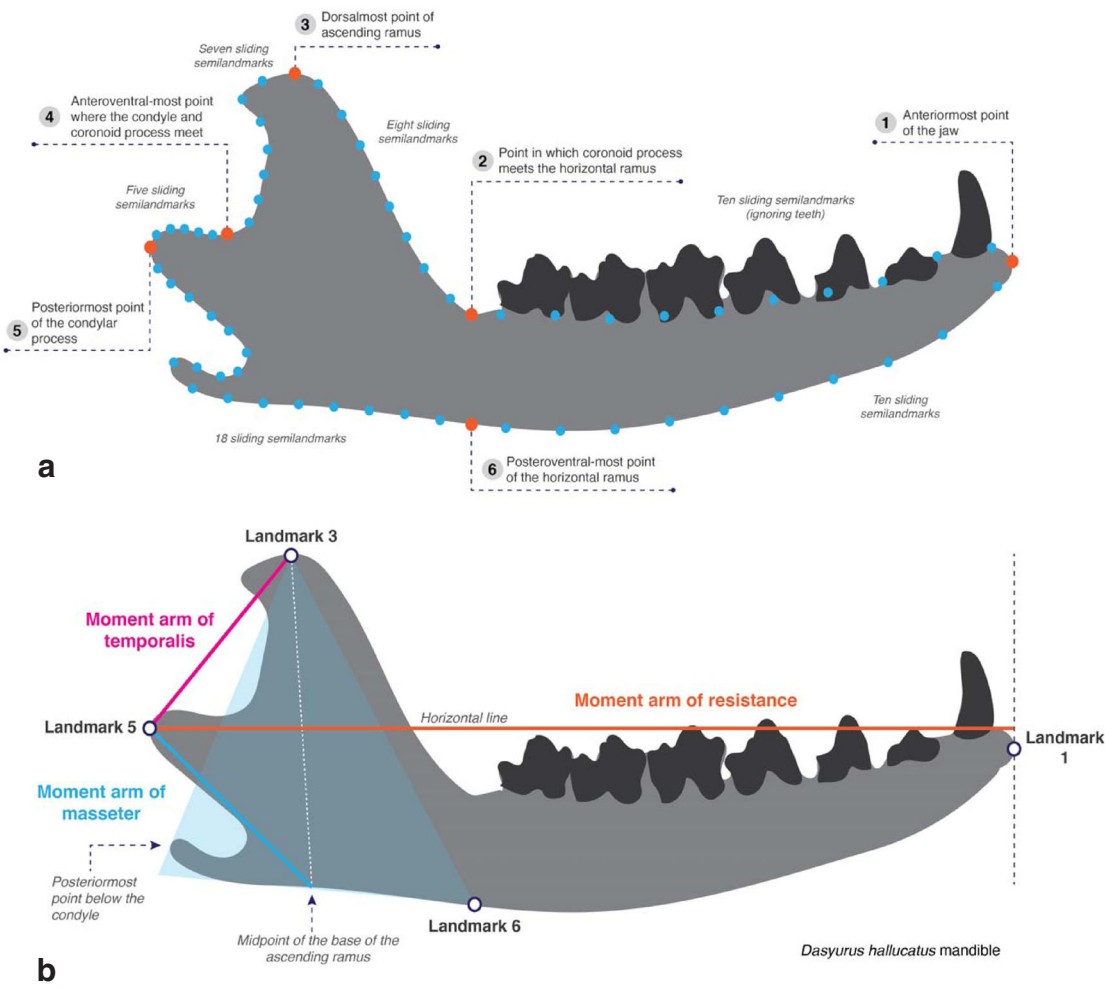

**Fig. 2 Data acquired from the jaws of Mesozoic and extant small mammals. a** Jaw landmarking regimen used in this study. Modified from ref. [12]. In orange: six fixed landmarks; in blue: 58 sliding semi landmarks. **b** Moment arm measurements taken in this study. Modified from ref. [19].

*Vincelestes* (#29) has previously been considered a carnivore on the basis of jaw shape[12]. Here, it plots near the morphospaces of both omnivores and herbivores. Bonaparte[24] considered the incisor wear of *Vincelestes* reminiscent of Cenozoic carnivores, and Rougier[25] considered its jaw morphology indicative of a forceful bite enabling the incorporation of tough plant matter into a primarily carnivorous/insectivorous diet.

*Therian crown mammals.* Extant marsupials have a large diversity of diets, including herbivory, but the extinct metatherians in our sample are considered to have been limited in diet to insectivory and carnivory (note that there are some putatively herbivorous/omnivorous extinct metatherians, like *Glasbius* and polydolopimorphians[39,40]). Their jaw shape is very similar to that of extant carnivores and insectivores (Fig. 3). Dental morphology indicates that *Eodelphis* (#32) and *Deltatheridium* (#30) were carnivorous, *Didelphodon* (#31) durophagous or molluscivorous[31,32], and *Alphadon* (#33) is considered to have been insectivorous, on the basis of its jaw shape and body size[12]. Dental microwear indicates a broad diet consisting of vertebrates, plants, and hard-shelled invertebrates for *Didelphodon*; biomechanical analyses of its skull and jaw points towards a durophagous diet[15,16]. Biomechanical analyses of the resistance to bending and torsion of *Eodelphis* jaws, points to a durophagous diet in *Eodelphis cutleri* and non-durophagous faunivory for *Eodelphis browni*[16]. Here, *Eodelphis*, *Deltatheridium* and *Didelphodon* plot closely to the extant carnivores, while *Alphadon* plots closely to the extant insectivores.

Extant placentals also have a wide range of diets, but many of the extinct eutherians in this study (i.e., *Sinodelphys* [#34], *Juramaia* [#35], *Eomaia* [#36], *Kennalestes* [#40], *Barunlestes* [#44], and *Kulbeckia* [#43]) are considered insectivorous[7,12]. Here, we corroborate this hypothesis (Fig. 3): all extinct eutherians plot within the insectivore morphospace, with the exception of *Asioryctes* (#38) which plots in the insectivore/carnivore morphospace, and *Juramaia* and *Sinodelphys*, which plot just outside the insectivore morphospace.

**Using jaw shape to infer diet in Mesozoic mammals.** We performed a phylogenetic flexible discriminant analysis (phylo FDA) following Motani and Schmitz[41] to determine the posterior probability of the Mesozoic taxa belonging to one of three dietary categories: insectivore, carnivore, or herbivore (we omitted omnivores as they are not well discriminated in Fig. 3). We used the first seven PC scores (of the PCA of Procrustes coordinates of jaw shape), which together accounted for 81.39% of the variance. The results of the analysis can be seen in Fig. 4 and the posterior probability values can be seen in Supplementary Data 1. We used the extant taxa of known diets as the training dataset for the discriminant analysis: these taxa were classified correctly 89.19% of the time. For the most part, we see a good separation between dietary groups among extant mammals (Fig. 4a), with some exceptions: the primarily herbivorous olingo (*Bassaricyon gabbii*, #94) plots with the carnivores (although mainly frugivorous, it

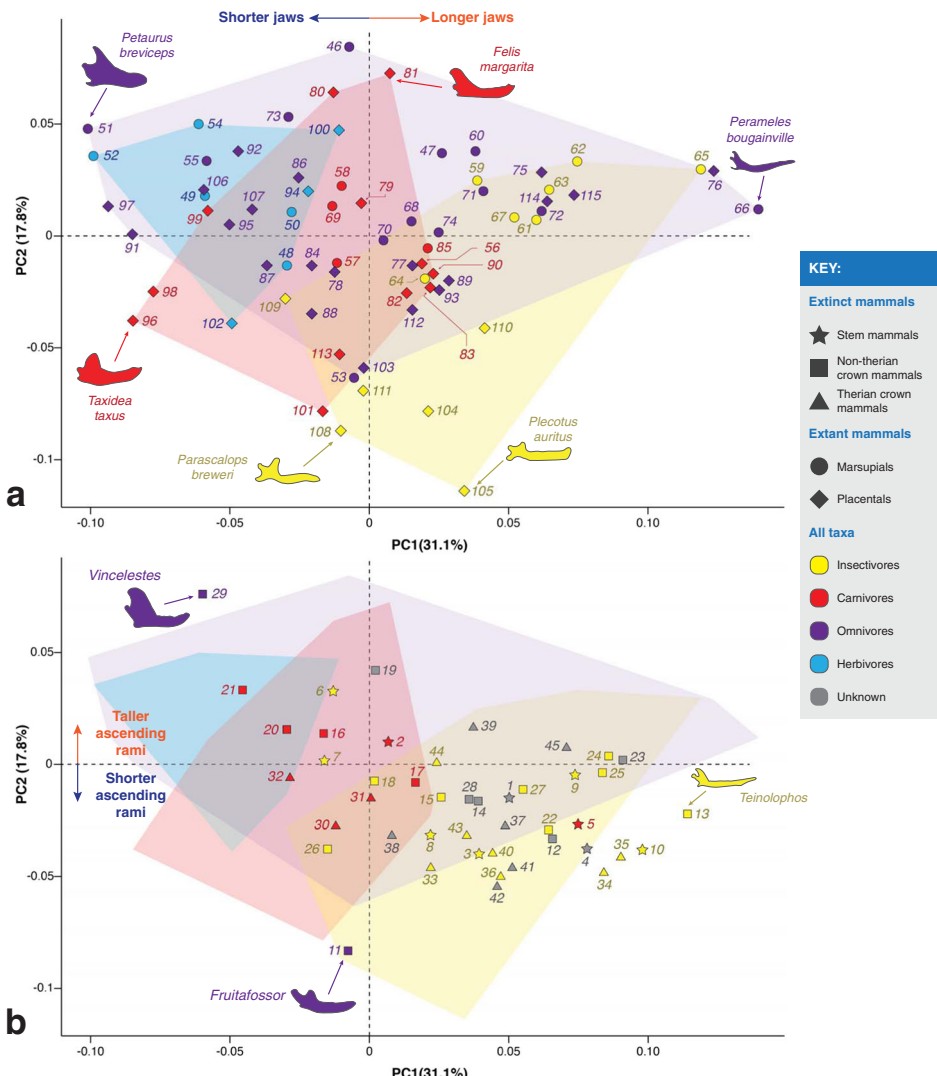

**Fig. 3 Scatter plots of the principal component analysis (PCA) results (PC1 vs. PC2). a** Extant taxa, **b** extinct taxa. Convex hulls shown for extant insectivores (yellow), carnivores (red), omnivores (purple) and herbivores (blue). Icon colors indicate known dietary categories of extant mammals and suggested dietary categories for Mesozoic mammals (obtained from the literature). See Table 1 for taxon names.

**Table 2 Summary of the Procrustes ANOVA (Type II, Conditional SS) performed for jaw shape data as a function of dietary group.**

| Statistic | Carnivore vs. | | | Herbivore vs. | | Insectivore vs. |
|---|---|---|---|---|---|---|
| | **Herbivore** | **Insectivore** | **Omnivore** | **Insectivore** | **Omnivore** | **Omnivore** |
| $R^2$ | 0.15795 | 0.1455 | 0.03417 | 0.25987 | 0.06404 | 0.06499 |
| $F$ | 4.1266 | 4.5973 | 1.6628 | 6.6712 | 2.6686 | 3.0582 |
| $Z$ | 3.0431 | 3.0556 | 1.2966 | 3.4824 | 2.1976 | 2.4708 |
| $p$ | **0.003** | **0.001** | 0.098 | **0.001** | **0.017** | **0.005** |

Significant $p$ values (<0.05) in bold letters. $n$ = 8 herbivores, 16 carnivores, 13 insectivores, 33 omnivores.

can consume small vertebrates), and a couple of insectivores plot very near the carnivores (i.e., the little brown bat [*Myotis luci-fugus*, #104] and the Hispaniolan solenodon [*Solenodon para-doxus*, #109]). These three taxa, alongside the carnivorous greater bulldog bat (*Noctilio leporinus*, #101), were the only extant taxa misclassified by the discriminant analysis.

The Mesozoic mammals included in our sample have largely been considered faunivorous and the results of the phylo FDA (Fig. 4b) corroborate this hypothesis. The majority of them are

classified as insectivorous, including most stem mammals, australophenidans, "symmetrodontans" and eutherians, among others. Among the eutriconodontans, *Argentoconodon*, *Gobico-nodon*, *Repenomamus*, and *Trioracodon*, are classified as carnivores, *Triconodon* and *Yanoconodon* are classified as insectivores, but with moderate support (posterior probabilities: 48% and 52%, respectively), and *Phascolotherium* and *Volati-cotherium* are more confidently classified as insectivores (poster-ior probabilities: 60% and 73%, respectively). Among the

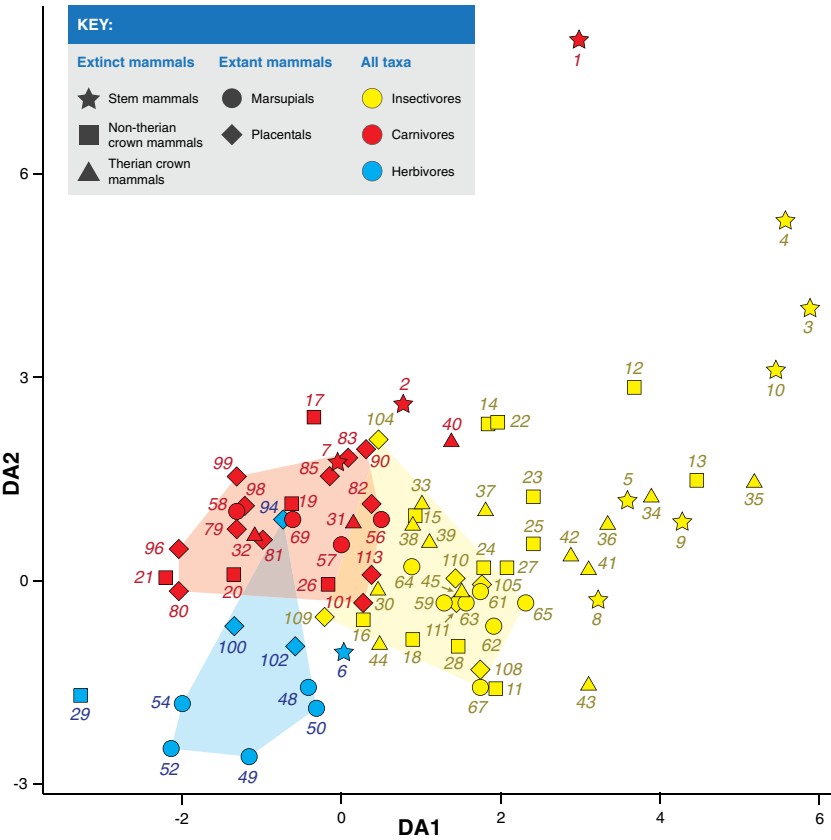

**Fig. 4 Phylogenetic flexible discriminant analysis results, showing discriminant axis 1 (DA1) and two (DA2), of all taxa in this study.** Extinct taxa are color coded based on their posterior probability of belonging to one of the established dietary categories. Convex hulls show the position of the extant taxa in the plot and are color coded based on their dietary categories.

metatherians, *Didelphodon* and *Eodelphis* are classified as carnivores, while *Alphadon* and *Deltatheridium* are classified as insectivores with moderate support (posterior probabilities: 54% and 52%, respectively). The stem mammals, *Haramiyavia*, *Sinoconodon*, and *Docofossor* are all confidently classified as carnivores (posterior probabilities over 80%), and the crown mammals *Crusafontia* and *Kennalestes* are also classified as carnivores, but with moderate support (posterior probabilities: 54% and 52%, respectively). Two taxa in the analysis are classified as herbivores, because of their relatively tall ascending rami: *Vincelestes* (#29) and *Haldanodon* (#6). The dental morphology of *Vincelestes* points to a primarily faunivorous diet[24], but it has been previously noted that its jaw morphology is indicative of a forceful bite; Rougier[25] suggested that this jaw morphology might have enabled *Vincelestes* to incorporate tough plant matter into its diet, but it might also be indicative of durophagy. The dental morphology[27] and body size of *Haldanodon* point towards an insectivorous diet; in this analysis, the posterior probability of *Haldanodon* being a herbivore is not high (only 40.3%). The evidence thus far suggests *Haldanodon* had a faunivorous diet; its jaw morphology might be indicative of the incorporation of tougher food sources into its diet.

**Mechanical advantage of the jaws of small mammals**. We obtained mechanical advantage (MA) data to test whether extant mammals of different dietary groups have distinct MA values (Table 3). The mechanical advantage measurements were standardized across all jaws to account for differences in jaw morphology (e.g., presence or absence of the angular process) (Fig. 2b); the outlever was measured at the anterior end of the jaw and at the first lower molar (m1). When measuring mechanical

advantage at the jaw tip and considering extant taxa only, we find statistically significant differences in the mechanical advantage of the masseter (MAM) values in all pairwise dietary combinations except for carnivore-insectivore (Table 3). The mechanical advantage of the temporalis (MAT) is statistically distinct only between herbivores and insectivores, and carnivores and insectivores (Table 3). Herbivores and carnivores do not have statistically distinct MAT values. This may differ in a sample of larger (> 5 kg) therians. When measuring the outlever at the m1, we find statistically significant differences in all pairwise comparisons of MAM between dietary groups, except for herbivore–omnivore and carnivore–insectivore. When considering MAT, we only find significant differences between omnivores and carnivores, insectivores and herbivores, and insectivores and carnivores.

Figure 5 shows the mechanical advantage of the masseter (left) and temporalis (right), measured at the jaw tip, in a phylogenetic context (see also Supplementary Fig. 7 for individual taxon names). Phylogeny appears to have a large influence on the mechanical advantage and diet of the jaws of small mammals. Most Mesozoic taxa have low (blue) to intermediate (green) MAM values. Most stem mammals have intermediate (green) to high (red) MAM values and non-therian crown mammals have low MAM values, with the exception of *Fruitafossor* and *Vincelestes* (the latter has the highest MAM value of all taxa, both extinct and extant). Most eutherians, both extinct and extant, have intermediate to low MAM values, with the exception of the relatively high values (yellow to orange) seen in elephant shrews (order Macroscelidea) and the four-toed hedgehog (order Eulipotyphla, *Atelerix albiventris*). Some members of the orders Carnivora (including canids and euplerids) and Afrosoricida have some of the lowest MAM values. Metatherians have MAM values

**Table 3 Pairwise *p* values (uncorrected significance) of one-way PERMANOVAs of the mechanical advantage values of the masseter (MAM) and temporalis (MAT) obtained in this study on extant taxa of known dietary preferences only (permutation *N* = 9999).**

| Measured at jaw tip | | | | | Measured at m1 | | | | |
|---|---|---|---|---|---|---|---|---|---|
| | MAM | | | | | MAM | | | |
| *F* = 6.664 | IN | HR | OM | CA | *F* = 6.813 | IN | HR | OM | CA |
| IN | | | | | IN | | | | |
| HR | **0.001** | | | | HR | **0.0011** | | | |
| OM | **0.0295** | **0.04** | | | OM | **0.0008** | 0.189 | | |
| CA | 0.7612 | **0.0006** | **0.0309** | | CA | 0.0603 | **0.0093** | 0.0817 | |
| *F* = 3.314 | MAT | | | | *F* = 3.817 | MAT | | | |
| IN | | | | | IN | | | | |
| HR | **0.0045** | | | | HR | **0.0197** | | | |
| OM | 0.2725 | 0.0847 | | | OM | 0.1682 | 0.3622 | | |
| CA | **0.0048** | 0.4721 | 0.1139 | | CA | **0.0022** | 0.3922 | **0.0376** | |

Outlevers for mechanical advantage calculations measured at the jaw tip (left) and first lower molar (m1, right). Significant *p* values (<0.05) in bold letters. *n* = 8 herbivores, 16 carnivores, 13 insectivores, 33 omnivores.
*IN* insectivore, *HR* herbivore, *OM* omnivore, *CA* carnivore.

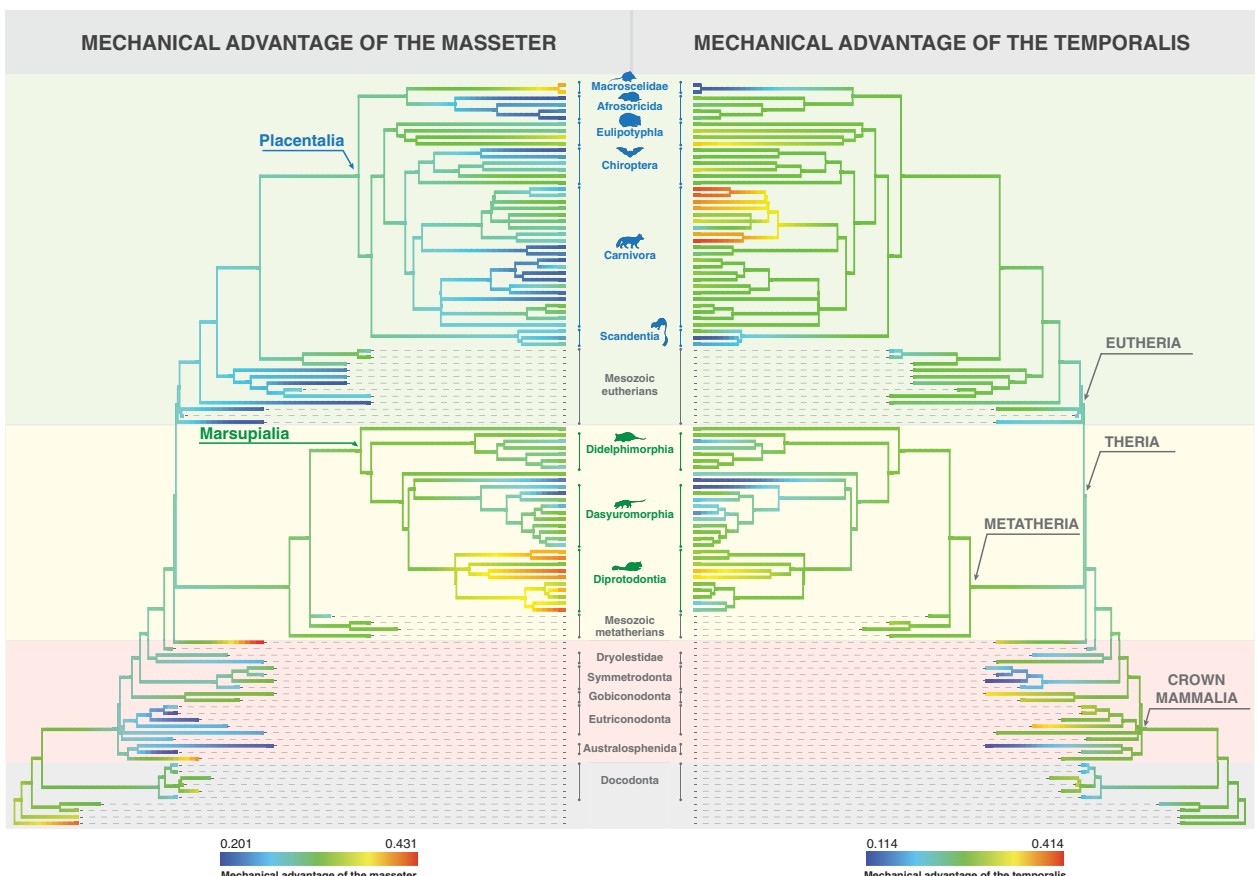

**Fig. 5 Mechanical advantage values of the masseter and temporalis when biting at the jaw tip visualized in the context of the phylogeny used in this study.** See Supplementary Fig. 7 for individual taxon names.

ranging from low to intermediate (in the orders Dasyuromorphia and Didelphimorphia, as well as in the Mesozoic metatherians) to some of the highest in the order Diprotodontia (e.g., the sugar glider [*Petaurus breviceps*], the woylie [*Bettongia penicillata*], the cuscus [*Phalanger orientalis*]).

Most taxa have intermediate MAT values (Fig. 5 and Supplementary Fig. 7). Very low MAT values are seen in the extinct non-therian crown mammals *Teinolophos* and *Zhangheotherium* and a few extant taxa, including marsupials like the

Western barred bandicoot (*Perameles bougainville*) and the numbat (*Myrmecobius fasciatus*), and placentals such as the striped treeshrew (*Tupaia dorsalis*) and the short-snouted elephant shrew (*Elephantulus brachyrhynchus*). The highest MAT values belong to members of the order Carnivora, including skunks (*Mephitis macroura* and *Conepatus humboldtii*), the least weasel (*Mustela nivalis*) and the tayra (*Eira barbara*). Some diprotodontians like the common ringtail possum (*Pseudocheirus peregrinus*) and the sugar glider (*Petaurus breviceps*) also have

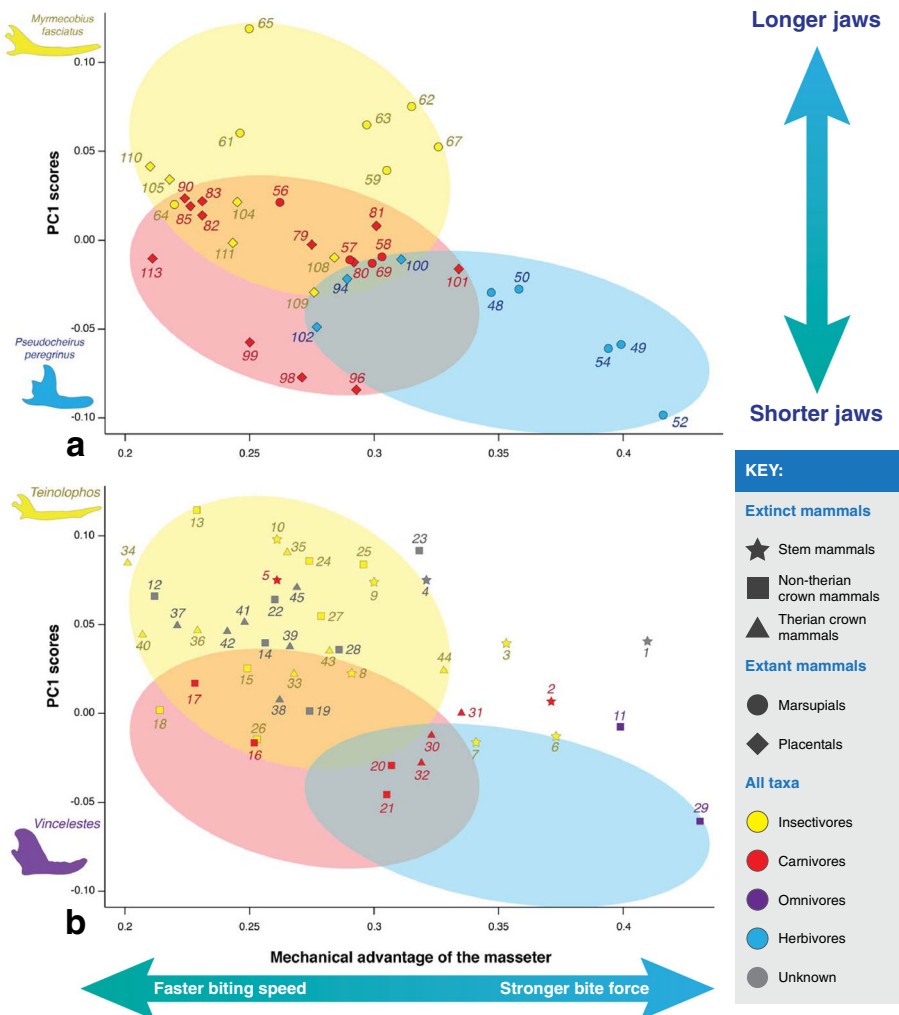

**Fig. 6 Scatter plot of the mechanical advantage of the masseter (*x* axis) vs. PC1 scores from Fig. 3 (*y* axis), which mainly describes jaw length. a** Extant taxa, **b** extinct taxa. Colors indicate known dietary categories of extant mammals and suggested dietary categories for Mesozoic mammals (obtained from the literature). Ovals indicate where extant taxa of known dietary categories plot, as in part **a**.

relatively high MAT values. Some extinct taxa also have relatively high MAT values, including the stem mammal *Docofossor*, and the non-therian crown mammals, *Triconodon* and *Vincelestes*.

Figures 6 and 7 present a visualisation of the mechanical advantage of the masseter and the temporalis (*x* axis, outlever measured at the jaw tip) and the PC1 scores of Fig. 3 (*y* axis, mainly describes jaw length) because, as previously mentioned, this is the axis in which dietary categories among extant mammals are best discriminated. In the *y* axis of Figs. 6a and 7a, herbivores have short jaws, carnivores have short to intermediate-length jaws and insectivores have intermediate-length to long jaws. In Fig. 6a, insectivores and carnivores have low mechanical advantage values of the masseter (i.e., when biting: less forcefulness, more speed), and herbivores have higher mechanical advantage values (i.e., when biting: more forcefulness, less speed). In Fig. 7a, insectivores have lower mechanical advantage values of the temporalis, while carnivores and herbivores have higher mechanical advantage values. Note that most carnivores have intermediate MAT values, but some mustelids (i.e., the least weasel [*Mustela nivalis*, #99], the American badger [*Taxidea taxus*, #96], and the North American river otter [*Lontra canadensis*, #98]), have the highest MAT values among extant mammals. Also note that, among insectivores, those with the highest MAT values are burrowing

vermivores (i.e., the short-tailed shrew tenrec [*Microgale brevicaudata*, #111], the hairy-tailed mole [*Parascalops breweri*, #108], and the Hispaniolan solenodon [*Solenodon paradoxus*, #109]). By using a combination of their MAM and MAT values (as well as their jaw length), we can distinguish dietary categories among extant mammals. We decided to omit omnivores from these figures because, as seen in Fig. 3, they cannot be distinguished from other dietary groups on the basis of jaw shape.

We also obtained additional mechanical advantage measurements, in which the outlever was measured at the first lower molar (m1), rather than the jaw tip (Supplementary Figs. 8, 10, 11, and 13). We made this alternative measurement because Grossnickle[17] found that the length of the posterior portion of the jaw (measured from the jaw joint to the m1) is a strong predictor of diet in mammals. Compared to the mechanical advantage (MA) measurements at the jaw tip (Figs. 6a and 7a), we see a less clear distinction between dietary groups among extant mammals. There is considerable overlap between dietary groups in Supplementary Fig. 10 (jaw length~MAM). In Supplementary Fig. 11 (jaw length~MAT), there is a better separation between dietary groups.

Based on previous likely determinations of diet of Mesozoic mammals (see Supplementary Data 1 for the full list of sources), most taxa plot where it is "expected" of them, with some

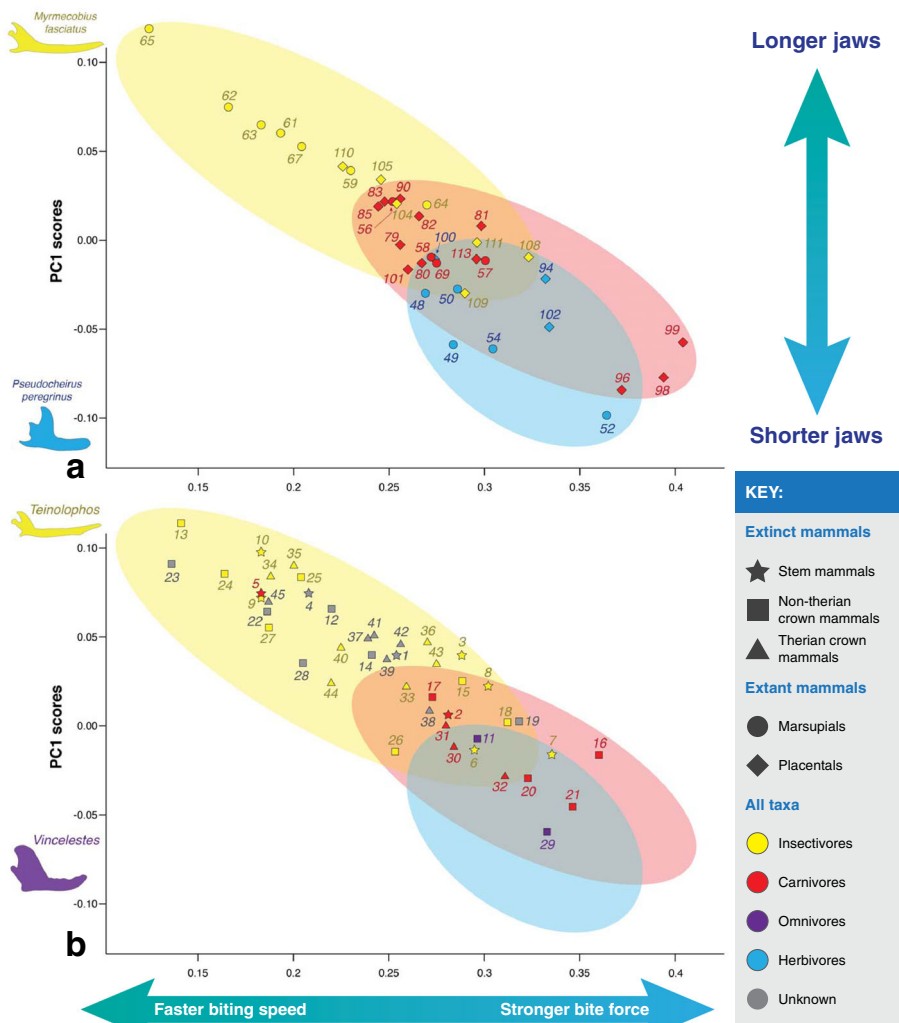

**Fig. 7 Scatter plot of the mechanical advantage of the temporalis (x axis) vs. PC1 scores from Fig. 3 (y axis), which mainly describes jaw length.**
**a** Extant taxa, **b** extinct taxa. Colors indicate known dietary categories of extant mammals and suggested dietary categories for Mesozoic mammals (obtained from the literature). Ovals indicate where extant taxa of known dietary categories plot, as in part **a**.

exceptions (Figs. 6b and 7b): 1) about half of the stem mammals (i.e., *Haramiyavia*, *Sinoconodon*, *Morganucodon*, *Haldanodon*, and *Docofossor*), most of which are thought to have been faunivorous, have higher MAM values than modern insectivores and carnivores, and 2) the docodontan *Castorocauda* has MAM and MAT values consistent with an insectivorous diet, as opposed to the carnivorous diet proposed for this taxon[7,28,33]. Most Mesozoic mammals have mechanical advantage values similar to modern insectivores, a few taxa are similar to carnivores (e.g., *Sinoconodon*, *Triconodon*, *Trioracodon*, *Argentoconodon*, *Gobiconodon*, *Repenomamus*, *Deltatheridium*, *Didelphodon*, and *Eodelphis*), and some are more similar to herbivores (e.g., *Vincelestes* and *Fruitafossor*).

## Discussion

In this study, we found that jaw shape is largely indicative of dietary category (with the exclusion of omnivores) in small extant mammals. In terms of length, herbivores have short jaws, insectivores long jaws and carnivores range from short to intermediate-length jaws. This jaw shape discrimination between dietary categories reflects the findings of Grossnickle and Polly[12]; however, we find a clearer distinction between carnivores and insectivores, which might be related to our larger sample size and

landmarking regimen. In Fig. 3 there is some overlap between these two dietary categories but, in general, hypercarnivores such as the felids (#79,80,81) and the Tasmanian devil (*Sarcophilus harrisi*, #58) are clearly distinguished from the insectivores, while some mesocarnivores like the quolls (genus *Dasyurus*, #56,57) and the Corsac fox (*Vulpes corsac*, #90) are more similar to the insectivores. This is in agreement with the study by Prevosti et al.[42] who also found a clear separation of hypercarnivorous mammals from mesocarnivores and insectivores. As hypothesized by Prevosti et al.,[42] and previous authors, hypercarnivores have shorter jaws to increase the mechanical advantage of the adductor musculature and deliver a stronger bite; alongside other morphological features, this configuration proves advantageous for prey subduing and meat consumption.

Generally, it is expected that herbivores would have a high mechanical advantage (i.e., increased bite force) of the masseter (MAM), as they need strong bites at low gapes, and that carnivores would have a high mechanical advantage of the temporalis (MAT), as they need strong bites at wide gapes (see ref. [43] and references therein). Interestingly, we found that herbivores not only tend to have high MAM, but also high MAT, while carnivores have high MAT, but low MAM. A study on cranial morphology of rodents[44] found a similar pattern: herbivores have enlarged masseter and temporalis muscles, while carnivores have

an enlarged temporalis and a reduced masseter. Despite their different diets, the enlarged temporalis muscle in herbivores and carnivores might serve a similar function: to resist dislocating forces when biting at the front of the jaw, either when dealing with struggling prey (in carnivores) or biting hard plant material (in herbivores)[44]. On the other hand, insectivores have lower mechanical advantage (i.e., increased biting speed) in both muscles, which would be beneficial for catching fast moving prey. This low mechanical advantage is a byproduct of their long snouts, advantageous for both speedy jaw closure and for some foraging strategies (e.g., capturing prey inside holes or burrows). Grossnickle[17] posited that faunivorous taxa benefit from having a shorter masseter (which attaches closer to the jaw joint than in herbivores) as it reduces the length of the in-lever (consequently lowering the mechanical advantage of the masseter) and allows them to have a wider gape. This is reflected in a shorter jaw joint to angular process distance in faunivorous taxa than in herbivores[17].

Having explored how jaw shape and mechanical advantage relate to diet in small extant mammals, we can evaluate whether these morphometric and functional metrics are good proxies for diet in Mesozoic mammals. Overall, we found good correspondence between jaw shape, mechanical advantage and diet in Mesozoic mammals. We corroborate the hypothesis that most Mesozoic taxa were insectivorous and some clades, like eutriconodontans, had a carnivorous diet. In Fig. 6, we see that many stem mammals have higher masseter mechanical advantage values than "expected" for their proposed dietary categories (also seen in Fig. 5). Stem mammals have a very anteriorly positioned angle of the mandible (note that the angular process of stem mammals may not be homologous to that of therians [see ref. 22 and references therein]). This is related to the presence of retained postdentary bones (located posteriorly to the angle) that are incorporated into an enclosed middle ear to a greater or lesser extent in more derived mammals[45]. The anterior position of the angle in turn increases the moment arm of the masseter. In contrast, non-therian crown mammals that have not attained a fully-enclosed middle ear, such as *Yanoconodon* (#15) and *Maotherium* (#25), do not have longer masseter moment arms because their mandibles lack an anteriorly positioned angular process, possibly related to the fact that the middle ear ossicles have now become medially separated from the dentary[45]. Therian mammals with a fully enclosed middle ear do not have anteriorly positioned angular processes.

In conclusion, we analyzed how jaw shape and mechanical advantage of the jaw adductor muscles of small extant mammals relate to their diets; this information was then used to infer dietary categories in Mesozoic mammals. Jaw shape alone can be used as a good indicator to roughly distinguish among herbivores, carnivores and insectivores, but cannot distinguish omnivores. Overall, this holds true for most Mesozoic mammals. When mechanical advantage values of both the masseter and temporalis are considered together, dietary categories can also be distinguished. This holds true for most Mesozoic mammals (note that the anterior position of the angular process in stem mammals confers them higher MAM values than expected for their diets). When we put all this information together, we find the following: herbivores have high MAM and MAT values, tall ascending rami and shorter jaws; carnivores have low MAM values, medium to high MAT values, medium to short ascending rami and jaw length; and insectivores have low to medium MAM and MAT values, short ascending rami and longer jaws.

These morphological and functional characteristics reflect the differential need of these animals for acquiring and processing particular types of food, which can be aided by having either stronger or faster jaw closure. Other factors, such as different foraging strategies and phylogenetic history, also play a role in determining the morphological configuration and functional traits of the mammalian jaws. The fact that the jaw shape and mechanical advantage of extant mammals could be successfully used as the basis for inferring the diet of mammals living during the Mesozoic (even those with retained postdentary bones), highlights that ecological pressures and jaw biomechanical principles were similar today and in the past.

We corroborate the hypothesis that most non-allotherian Mesozoic mammals had a faunivorous diet. Considering the taxa in our sample, most stem mammals, "symmetrodontans", dryolestids, amphitheriids and eutherians appear to have had an insectivorous diet (or one consisting of "soft" aquatic invertebrates in the case of mammals proposed to have been semi-aquatic, such as *Castorocauda* and *Teinolophos*), while the eutriconodontans and metatherians studied here probably had a more carnivorous diet. Fossorial or semi-fossorial mammals with a potentially vermivorous diet can also be distinguished from other insectivores as their jaw shape is more similar to that of carnivores.

## Methods

For a diagram of the methods used in this paper, refer to Supplementary Fig. 1.

**Materials**. We used photographs of the jaws of 70 small extant mammals and 45 extinct Mesozoic mammalian taxa spanning the Late Triassic to the Late Cretaceous. The full list of taxa can be seen in Table 1. The extant taxa chosen for this study (29 marsupials in six orders, 41 placentals in six orders) are based on those used in Grossnickle and Polly[12]; new taxa were added for greater taxonomic and dietary diversity. Following ref. 3, most extant mammals in our sample weigh under 5 kg as most Mesozoic mammals were under this body mass threshold. Five extant species over 5 kg were included in the sample (i.e., *Sarcophilus harrisii*, *Lynx rufus*, *Lontra canadensis*, *Taxidea taxus*, and *Procyon lotor*) to reflect the upper body size limit of larger Mesozoic mammals such as *Repenomamus* (12–14 kg[4]).

For this study, we chose to exclude the Mesozoic haramiyidans and multituberculates, and only to focus on non-allotherian Mesozoic taxa. Grossnickle and Polly[12] had previously determined that the jaw shape of multituberculates is different from other non-allotherian Mesozoic mammals. We attempted to include allotherians of different diets in the sample, but they all plotted in their own area of morphospace, far away from other non-allotherian Mesozoic mammals. They were also dissimilar to any extant mammals in our sample (Supplementary Fig. 6). Compared to non-allotherian Mesozoic mammals and extant mammals of different diets, multituberculates and haramiyidans had higher mechanical advantage values, which skewed our data heavily (Supplementary Fig. 9). Allotherians have a very derived jaw morphology, dissimilar to other Mesozoic mammals and extant small mammals; they also had palinal jaw movements, unlike any other extant or extinct mammal[46], which could lead to a unique biomechanical biting performance. Therefore, we decided to exclude allotherians from this sample, with the exception of *Haramiyavia*, which may not be closely related to later Jurassic euharamiyidans, following ref. 47. Additionally, early haramiyidans (like *Haramiyavia*), might have not had the palinal jaw movements of later allotherians[48].

Photographs of extant mammal jaws were obtained from the online databases: Animal Diversity Web (ADW) of the University of Michigan[49] (https://animaldiversity.org), the Natural History Museum (NHM) online database (https://data.nhm.ac.uk) and the Field Museum online database (https://collections-zoology.fieldmuseum.org). For a detailed list of the extant mammal specimens used in this study refer to Supplementary Data 1. All photographs were reviewed by NMMG to ensure the jaws were all captured in the same orientation; some photographs were rotated in order to have the horizontal ramus of the jaw parallel to the *x* axis.

The extinct taxa considered in this study include ten stem mammals, 19 non-therian crown mammals, and 16 therian crown mammals. Photographs were primarily obtained from the literature[1,2,5,15,23,28,33,35,36,38,50–59]. Additionally, photographs were taken from specimens held at the Institute of Paleobiology, Polish Academy of Sciences (Warsaw, Poland), at the Oxford University Museum of Natural History (Oxford, United Kingdom), at the Natural History Museum (London, United Kingdom), and at the Steinmann Institut, Universität Bonn (Bonn, Germany). Photographs were taken by NMMG. For a full list of the literature and museum collections used to source these photographs refer to Supplementary Data 1.

Dietary information for extant taxa was obtained from the Animal Diversity Web[49]. Proposed dietary preferences for Mesozoic mammals were obtained from the literature. The full list of taxa, their dietary preferences, and the detailed sources of this information can be seen in the Supplementary Data 1. While no Mesozoic mammals specialized for herbivory (i.e., some multituberculates and haramiyidans)

were included in our sample, we decided to include some extant herbivores whose jaw morphology is not as derived as that of rodents for comparative purposes.

**Phylogenetic information**. We built a phylogeny in Mesquite incorporating all the taxa used in this study (see the Supplementary Figs. 2–5 for details). The overall topology of the phylogeny of the Mesozoic taxa was from refs. [60,61]; additional sources were used to refine the position of *Haramiyavia*[47], and the phylogenetic relationships within Morganucodonta[33] Docodonta[36], Australosphenida[52], Eutriconodonta[5,54], 'Symmetrodonta'[55], Dryolestidae[62], Metatheria[63], and Eutheria[59,64–66]. Similarly, the overall topologies of the Placentalia and Marsupialia phylogenies were obtained from refs. [67,68], respectively. Supporting literature was needed to account for all taxa included within Scandentia[69], Carnivora[70], Chiroptera[71], Eulipotyphla[72], and Afrosoricida[73].

The phylogeny was time-scaled using the "equal" method of Brusatte et al.,[74] using the package "paleotree" 3.3.0[75] in RStudio 1.2.1335 (RStudio team). Appearance dates for extinct taxa were obtained from the Paleobiology Database (http://fossilworks.org) and ref. [52]. Divergence dates to constrain the nodes were obtained from a diversity of phylogenies of Mesozoic taxa[47,60,61], Marsupialia[68], Scandentia[69], Carnivora[70], Chiroptera[71], Eulipotyphla[72], Afrosoricida[76], and Macroscelidea[77].

**Geometric morphometrics**. We performed a 2D geometric morphometrics study using fixed landmarks and sliding semi-landmarks in the jaws of small extant and extinct Mesozoic mammals. We used the same fixed landmarks as Grossnickle and Polly[12], with the exception of landmark 7 (i.e., posteroventral-most point of the angular process). We removed this from our analysis because many taxa in our study did not have an angular process. Additionally, we incorporated 58 sliding semi landmarks as seen in Supplementary Fig. 6a. All jaws were landmarked using TPS software by Rohlf[78]: tpsUtil was used in the construction of a file containing all images to landmark, tpsDig was used to digitize landmarks and semi-landmarks, and tpsRELW was used for Procrustes alignment. The resulting Procrustes aligned landmark coordinate data of the extant data were submitted to a principal components analysis (PCA) in RStudio, using the package "geomorph"[79]; convex hulls were drawn to indicate the morphospace occupation of the different dietary categories. The Mesozoic mammal jaw shape data was then projected onto the extant mammal morphospace by multiplying their Procrustes aligned landmark coordinate data by the PC variable loadings of the extant taxa (i.e., PC rotation scores).

**Mechanical advantage**. We measured the moment arms of resistance at the m1 and at the anterior end of the jaw, as well as the moment arms of the temporalis and masseter muscles in ImageJ following Supplementary Fig. 6b (modified from ref. [19]); these measurements roughly estimate the moment arms of both superficial and deep heads of the adductor muscles of the jaw. The moment arm of the medial pterygoid muscle, although not considered here, is probably very similar to that of the masseter. We calculated the mechanical advantage of the adductor muscles as follows: moment arm of the muscle divided by the moment arm of resistance at the bite point (m1 or jaw tip). A limitation of this technique lies on its 2D approach: using this method, we can only calculate the moment arms for pitch rotation, while any three-dimensional movement of the jaw (i.e., jaw or roll) cannot be quantified (see ref. [22]). Additionally, these measurements assume that the position of the pitch axis of rotation is at the jaw joint in all taxa.

**Statistics and reproducibility**. Statistical analyses follow Navalón et al.,[80] who quantitatively tested the relationship between beak shape, mechanical advantage, and feeding ecology in modern birds. In order to test for significant differences in jaw shape (as represented by Procrustes coordinates) between dietary groups, Procrustes ANOVAs (Type II, Conditional SS) were run in R using the function procD.lm of the package geomorph 3.1.2[79]. Pairwise comparisons of extant taxa between different dietary groups were performed (i.e., carnivores vs. herbivores, carnivores vs. insectivores, carnivores vs. omnivores, herbivores vs. insectivores, herbivores vs. omnivores, and insectivores vs. omnivores), considering eight herbivores, 16 carnivores, 13 insectivores, and 33 omnivores. The previously generated time scaled phylogenetic tree was included in this analysis, and pruned on a case-by-case basis, to account for the phylogenetic relationships of the taxa considered here using the packages ape 5.3[81] and geiger 2.0.62[82]. $R^2$, F, Z, and p values are reported in Table 2. The code used to perform Procrustes ANOVAs can be found in the "Code availability" section.

One way PERMANOVAs (permutation $N = 9999$) were run in PAST 3.24[83] to test for significant differences between dietary groups on the basis of the mechanical advantage values of their masseter and temporalis adductor muscles (outlevers measured at the jaw tip and m1). As above, pairwise comparisons of extant taxa between different dietary groups were performed, considering eight herbivores, 16 carnivores, 13 insectivores, and 33 omnivores. F and p values are reported in Table 3.

**Phylogenetic flexible discriminant analysis (phylo FDA)**. A phylo FDA was performed following ref. [41]. We performed this analysis to determine the posterior probability of the Mesozoic taxa of belonging to one of our established dietary categories (i.e., herbivore, carnivore, or insectivore), while considering their phylogenetic relationships. Extant omnivores were not included in this study because of their large dietary variability. The analysis was performed in R Studio 1.2.1335 using the packages ape 5.3[81], class 7.3-15[84], geiger 2.0.62[82], lattice 0.20-38[85], mda 0.4-10[86], nnet 7.3-12[84], using the source code (phylo.fda.v0.2.R) of ref. [41]. This analysis was performed by using the first 7 PC scores of the PCA of Procrustes coordinates of jaw shape, which together account for 81.39% of the variance; a lambda value of 0.08 was used. By using this configuration, 89.19% of the extant taxa were classified correctly.

**Data visualization**. Mechanical advantage values were plotted on a phylogeny of the taxa of interest using the package "phytools" version 0.6.99[87] in RStudio. Jaw shape and mechanical advantage were plotted together in a morphofunctional landscape in MATLAB R2019a 9.6.0 (The MathWorks, Inc., Natick, Massachusetts) following a protocol from Dr. J. A. Bright and previously used in Navalón et al.[80]. This visualization can be seen in the Supplementary Figs. 12 and 13, and is described in the associated text.

**Reporting summary**. Further information on research design is available in the Nature Research Reporting Summary linked to this article.

## Data availability

All data used in this paper is deposited at: https://data.bris.ac.uk/data/dataset/awok7xqxmjyg2kr1m6op92w8e[88]. It includes the data used for time scaling the phylogeny, for running the principal components analysis of jaw shape coordinates, for visualizing the mechanical advantage values on a phylogeny, for performing Procrustes ANOVAs, and for performing the phylogenetic flexible discriminant analysis. The file Supplementary Data 1 is a spreadsheet that includes the list of taxa used in this study, their PC scores, mechanical advantage values (measured at jaw tip and m1), observed diet (extant mammals), proposed diet (extinct taxa), phylo FDA results (i.e., discriminant axis scores, predicted dietary class, and probability of belonging to a dietary group), first and last appearance dates, and references (photographs, diet and first and last appearance dates). The sources of all the specimens analyzed here are described in detail in Supplementary Data 1; they include museum collections (Institute of Paleobiology, Polish Academy of Sciences, Warsaw, Poland; Oxford University Museum of Natural History, Oxford, United Kingdom; Natural History Museum, London, United Kingdom; Steinmann Institut, Universität Bonn, Bonn, Germany), online databases (Animal Diversity Web of the University of Michigan[49] [https://animaldiversity.org], the Natural History Museum online database [https://data.nhm.ac.uk] and the Field Museum online database [https://collections-zoology.fieldmuseum.org]) and photographs from the literature[1,2,5,15,23,28,33,35,36,38,50–59].

## Code availability

All the code used in this paper can be run in RStudio and can be found here: https://doi.org/10.5523/bris.awok7xqxmjyg2kr1m6op92w8e[88]. It includes the code used for time scaling the phylogeny, for running the principal components analysis of jaw shape coordinates, for visualizing the mechanical advantage values on a phylogeny, for performing Procrustes ANOVAs, and for performing the phylogenetic flexible discriminant analysis.

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

## Acknowledgements

N.M.M.G. is funded by a PhD scholarship (689870) from CONACYT (Consejo Nacional de Ciencia y Tecnología), México. P.G.G. and E.J.R. received funding from NERC grant NE/K01496X/1. N.M.M.G. would like to thank the Palaeontological Association for awarding me the "Stan Wood Award" from their Small Grants Scheme (PA-SW201701), which allowed me to visit several museum collections (Oxford University Museum of Natural History; Natural History Museum, London; Steinmann Institut, Bonn, Germany) in which I collected data relevant to this study. NMMG would like to thank Dr. Guillermo Navalón, Suresh Singh, and Dr. Armin Elsler for their help implementing the methods used in this paper. We would like to thank all the curators and researchers who gave us access and helped us photograph material in their collections: Dr. Łucja Fostowicz-Frelik, Dr. Sergi López-Torres, and Justyna Słowiak (Institute of Paleobiology, Polish Academy of Sciences), Dr. Hilary Ketchum and Katherine Child (Oxford University Museum of Natural History), Dr. Mike Day (Natural History Museum, London), Prof. Thomas Martin, Dr. Julia Schultz, and Dr. Kai Jäger (Steinmann-Institut, University of Bonn, Germany). We would like to thank Dr. David Grossnickle, Prof. Greg Wilson and an anonymous reviewer for their valuable comments that helped strengthen this study.

## Author contributions

N.M.M.G. co-designed the study, collected the data, ran the analyses, and wrote the manuscript. C.M.J., P.G.G., and E.J.R. contributed to the design of the study, supervised the project and commented on the manuscript.

## Competing interests

The authors declare no competing interests.
