## [Peer Review File · Communications Biology]

Reviewers' comments:

Reviewer #1 (Remarks to the Author):

See attached file. 
Reviewer #2 (Remarks to the Author):

The article entitled "Jaw shape and mechanical advantage indicate diet in Mesozoic crown, but not stem, mammals" by Morales-Garcia and co-authors tests whether jaw shape and jaw mechanical advantage are correlated and whether they are correlated to diet in small-bodied extant mammals and in turn in Mesozoic mammals. Ecomorphological studies of Mesozoic mammals have increased in recent years, focusing on various aspects of morphology and ecology (teeth and diet, jaws and diet, postcrania and locomotion, etc.). This study mainly builds on work by Grossnickle and Grossnickle and Polly. It largely uses the same landmarks and extant sample, but adds some methodological twists and some taxa both extant and extinct. It also compares estimates of mechanical advantage of the adductor muscles to shape as measured from GM. Overall, I thought that the paper was well written, the analytical approaches were sophisticated (but sometimes not clearly detailed), the direction of the paper has good potential, but I thought that it lacked depth of explanation and a major evolutionary punchline beyond the methodological implications. I don't know the journal but see that it's in the Nature family of journals. As it presently stands, I would not recommend it for publication in this journal, perhaps after major revision. My detailed comments are below:

- 1) The abstract should have more information in it? How many taxa sampled, which ones? Strikes me as rather vague, light on details.
- 2) The use of images from the literature should be cautiously approached. There was no discussion of how images were vetted. Images taken of jaws in lateral view can be at slightly different angles and this can affect measurements central to this paper. Was that evaluated? I didn't see mention of it.
- 3) The presentation of results can by nature be rather dry. However, rather than series of statements about correlations, I would encourage the authors to try to provide a topic sentence for each paragraph and have that topic sentence give the punchline of what will be detailed in the subsequent sentences. This allows the reader to trudge through the details with some sense of what is meaningful and what should be coming.
- 4) In the end the results indicate that in extant mammals the GM shape measures and mechanical advantage are significantly correlated to each other and with diet (MAT more than MAM). Many of the results are as you would expect. Notably, GM and mechanical advantage are measuring very similar components (some of the GM landmarks are the same position of MA measurements). So these aspects don't strike me as particularly novel. However, the discrimination of insectivores and carnivores is more novel, so why not spend some more time discussing those results. And that there is no statistical difference b/w herbivore MAT and carnivore MAT. I would encourage the authors to spend more time digging into these aspects of the results.
- 5) When it comes to the fossil data, why have the authors given more weight to pre-existing dietary inferences. Whose to say that those data are better estimates of the diets of these taxa. This is another area where I thought they should've provided greater discussion. What is the rationale for taking as "truth" the dental inferences or inferences from previous authors?
- 6) I couldn't tell from the described methods but it seemed as though they put extant and extinct data together and then ran the PCA. Instead, I would recommend that they run the PCA with only the extant data and then subsequently project the extinct data into that morphospace. In this recommended approach you're essentially predicting diet of the extinct taxa using the jaw shape space formed by extant taxa of known diets. As is, the extinct taxa are strongly influencing the shape space.

7) While they mention that stem mammals fall outside of the shape space occupied by extant taxa, in reality it looks as if many extinct taxa (not just stem mammals) fall outside of that shape space. I would've appreciated more discussion of what that means for functional inference. I would argue that the principles of biomechanics still apply. So how can we use these to still constrain dietary inference.

8) Ultimately, I felt that the paper could use further digestion of the results and further exploration into what these results mean more broadly for the evolution and paleoecology of Mesozoic mammals.

Reviewer #3 (Remarks to the Author):

Different morphological traits under selective pressures, like the tooth shape, linear measurements of postcranial elements, were widely used to investigate the ecomorphospace disparity of extinct groups in the geological history. The authors expanded a morphometric analysis of lower jaw, initiated by Grossnickle and Polly (2013), from linear measurements to geometric morphometrics, and made use of modern analogues, which to assess changes in mammalian morphological and dietary disparity. This geometric morphometrics may increase the applicability of statistical analysis in fossil study and reduce the possible artificial influence. The authors further used the functionally-driven approaches to clarify the relationship between the mechanical advantage and ecomorphospace disparity. The result shows some correlation between the shape and function, and the two traits have the potential to be the proxies to elucidate the dietary preferences of extinct mammals. I would recommend the manuscript for publication if the following points are addressed.

1. The study of jaw shape and diet lacks statistical correction for phylogenetic non-independence. The PC1 and PC2, especially PC2 for scatter plots of the Principal Component Analysis, are more or less related to the Phylogeny of the chosen taxa. When we look at the plots of all extinct taxa, most Carnivores, Insectivores, Omnivores are not fallen into the corresponding morphospace of extant taxa. But if we look at those plots fallen into the overlap zone of extant and extinct taxa, most of which belong to the Therian crown mammals, the diet of these taxa show a good congruence with those of extant mammals. Because the dietary categories are made by the data of extant therian mammals, the Therian crown mammals within the overlap zone conform to the PCA, where all taxa belong to therian mammals and no phylogenetic information among them. But obviously, the morphology of jaws is strongly correlated with diet. Like the diet fitness of the Mesozoic mammals are much better when the Phylogenetic Generalized Least Squares (PGLS) regression was used to test the relationships between jaw shape and mechanical advantage incorporating phylogenetic relationships and phylogenetic signal. Thus, I strongly suggest using a Phylogenetic principal component analysis to perform a PCA considering the phylogenetic non-independence between taxa, which may give us the true information for correlation of the jaw shape variation and diet for extinct mammals, and finally help to know that whether jaw shape could be used as a dietary proxy in Mesozoic mammals.

2. The results of the ANOVA analysis, such as the principal components, should be listed somewhere in the manuscript.

3. For the taxa sampling, I feel like if some representatives of multituberculates (especially a few taxa representing herbivores and Omnivores) and haramiyidans included in this exploration, the result would be more informative. For the extinct taxa, there is no herbivory representative in this analysis, and the authors excluded multituberculate because of their derived morphology, while multituberculates, especially the Late Cretaceous multituberculates, is one of the few mammalian groups that could be referred to herbivores in Mesozoic mammals. To be honest, the morphology of lower jaw in multituberculates did not change too much during their evolution. I would suggest use multituberculates to fill the vacancy of lacking herbivores data.

On the other hand, the authors claimed that the anteriorly positioned angle of the mandible is the

reason that stem mammals largely plot in a different area of morphospace to extant mammals. About the view of Haramiyidans, or Euharamiyidans from China, their controversy is mainly on the presence of postdentary bones or not. If the analysis contains representatives of two different kinds of the lower jaw from the same groups, a comparison could be used to test the authors' hypothesis of the stem mammals. Since the anteriorly positioned angle of the mandible and retained postdentary bones are the important difference between stem mammals with others, clarify this question is crucial for the usability of this method in stem mammals.

Minor issues:

- In my opinion, the description of Mesozoic mammal plotting in the Supplementary should be an important component of the Discussion. Using the Mesozoic mammal data to solve scientific problems will be of interest to most researchers. Instead, there are some verbose expressions in the Result and Discussion of the main text.
- The authors stated that "Mesozoic mammals are now considered to have been more ecologically diverse, although all remained small (< 5 kg) [1,2]" in the first paragraph (page 1), which is inaccuracy. Some taxon, like Repenomamus or Vintana, could have body mass larger than 5 kg.
- Typo in the legend of Supplementary Fig. 6: A) Extant taxa; B) Extinct taxa.

REVIEWER COMMENTS

Please find our response to the reviewer comments in green text below. In our revised manuscript, all changes to the main text are in blue type, except for text relating to the m1 outlever measurement study, which is in red type.

REVIEWER 1

Overview

The study by Morales-García and colleagues seems to be inspired in part by one of my papers, Grossnickle & Polly (2013). I'm proud of that project. But in hindsight, I now recognize some critical concerns with my jaw-diet correlation analysis (e.g. small sample of extant taxa, angular process issues, lack of phylogenetic correction). What I appreciate most about the study by Morales-García and colleagues is that they address (or learn from) many of the shortcomings of Grossnickle & Polly (2013). For example, they incorporate phylogeny into analyses, avoid using the angular process as a landmark, and merge functional properties (mechanical advantage) with morphology in their analyses. Thus, I find their study to be a thorough and interesting examination of early mammal jaws, diet, and biomechanics. It's a strong scientific contribution that presents novel results, and I support it for publication.

However, I do have a few major concerns that may require major revisions. I discuss these below. Additional comments are mostly minor suggestions.

Our reply: We really appreciate all the comments of Dr. Grossnickle to strengthen our paper. Below, we address all of his comments and we upload a revised version of the manuscript taking into consideration the remarks raised by the reviewers.

MAJOR CORRECTIONS

Inferring stem mammal diets

We know the diets of modern taxa, and therefore we can test for correlations between their jaws and diet. However, you also present your study as a test of how well *fossil* jaws correlate with diet (as even stated in the title of the paper). But unlike modern taxa, we don't have reliable dietary information for most fossils. You even state, "there is no clear consensus on the diets of many Mesozoic taxa typically considered as "generalized insectivores"."

So, if we don't have a good understanding of early mammal diets, then how can you conclude that jaw morphology and mechanical advantage are poor indicators of stem mammal diets? I somewhat disagree with the crux of the paper, including your title. I think your analyses simply present new or updated hypotheses on the diets of early mammals.

We have modified the focus of the paper and now use mechanical advantage and jaw shape as predictive tools for the diet of Mesozoic mammals- projecting the extinct taxa onto the extant taxa morphospace (as suggested by two reviewers) was particularly valuable for this

purpose. We still discuss previous hypothesis for the diet of Mesozoic mammals and compare them with the results of our study.

For stem mammals (especially docodontans), I think you're too confident in previous dietary inferences (treating them as truth), and not confident enough in your own results. You might have predicted docodont diets better than you realize. You have *Agilodocodon* classified as an omnivore, and describe it as a "plant-dominated omnivore" in the text. (You mistakenly cite OPC data from Chen et al. 2019 – see my comment below on this.) *Agilodocodon* was inferred to be an omnivore because we (Meng et al. 2015) thought the incisor morphology showed evidence of gumnivory. But Wible & Burrows (2016, *Palaeontologia Polonica*) refuted the gumnivory hypothesis, arguing that *Agilodocodon* is an insectivore. Your analyses support Wible & Burrows by showing evidence of insectivory, not omnivory. Thus, your results contribute to two competing hypotheses, rather than being an incorrect prediction of diet.

By projecting the fossil taxa into the morphospace, we are able to predict diet of fossil taxa. Hence, we have changed the text the following way (page 3, lines 75-80):

“Agilodocodon was previously considered to be a plant-dominated omnivore, with exudativorous dental features which pointed to a diet mainly composed of plant sap (Meng et al., 2015); more recently, Wible and Burrows (2016) contradicted this hypothesis and suggested that the teeth of *Agilodocodon* most closely resemble those of extant insectivores. Here, *Agilodocodon* plots firmly within the morphospace of extant insectivores, close to the insectivorous dusky antechinus (*Antechinus swainsonii*) and the elephant shrews (*Elephantulus rufescens* and *E. brachyrhynchus*), which are insect-dominated omnivores.”

You use *Docofossor* as another example of a taxon that doesn't match predictions, because it is more similar to extant carnivores, not insectivores (as predicted). But in looking at the PCA plot in Supp Figure 6, *Docofossor* has a PC1 score that still overlaps well with insectivores, aligning with the hairy-tailed mole, a fossorial insectivore. It looks like many of the insectivores on the left (negative PC1 space) probably have diets that are heavy on earthworms (*Solenodon*, tenrec, mole, shrew). And *Docofossor* was almost certainly fossorial (based on the extreme limb adaptations), making it very likely that it was also eating soft invertebrates like worms. I think the PC1 comparisons to extant taxa support this inference. Again, I think you're under-confident in your own results.

As mentioned above, projecting fossil taxa into the morphospace allows us to make prediction of fossil dietary ecology. As such, we have changed the text the following way (page 3, lines 88-94):

Docofossor has skeletal features indicative of a fossorial lifestyle and a dentition similar to those of modern mammals foraging underground, such as moles, solenodons, and tenrecs (Luo et al., 2015). This docodontan has previously been considered an insectivore. Here, *Docofossor* plots within the morphospace of extant carnivores; however, it plots close to the burrowing Hispaniolan solenodon (*Solenodon paradoxus*), which has an insectivorous diet. In fact, among the extant insectivores in our sample, the burrowing vermivores (e.g., hairy-tailed mole, *Parascalops breweri*, and the Hispaniolan solenodon) have more negative PC1

scores than other insectivores (similar to that of *Docofossor*), and their PC1 values are more similar to those of carnivores.

My suggestion is to re-frame the paper, with the central message being that the analyses are powerful tools for inferring diets in crown AND stem mammals (excluding allotherians), and your results corroborate many previous dietary hypotheses. The power to predict fossil diets probably weakens in taxa that are further from the therian node. And this is supported by your observation that many non-therian taxa are far from extant taxa in morphospace. So I can understand some discussion on why the diet inferences for stem mammals are less reliable than those for crown mammals. But I don't think this is a big enough issue to make it a central point of the paper or title.

Our reply: We agree with the reviewer on this point. We have re-framed the paper based on this point, as per our reply above.

Making a morphospace with extant and extinct taxa

Fossils comprise a large component of the overall sample for the PCA, so they're going to account for a lot of the variance in the data and influence the PC axes and morphospace plot. This might be problematic. If a goal of the study is to use modern taxa with known diets to assess the unknown diets of fossils, then it makes more sense to me to use a modern-only dataset to produce the PCA morphospace. And then you can 'project' the fossil taxa onto that morphospace to better compare their morphology to those of the extant taxa. For example, I projected the fossils onto the 'modern morphospace' plot in Grossnickle & Polly (2013). For another example of 'projecting' geometric morphometric (GMM) data, see Wilson (2013, Paleobiology). Wilson used TPS software, so it's more relevant to your methods. If you follow this suggestion, it'll likely lead to the fossils plotting closer to the extant taxa than they are in the current PCA plot. Similarly, you could run additional multivariate analyses like an LDA or CVA, and treat the fossils as 'unknowns' and see which extant dietary groups the fossils are classified into (e.g., see Chen & Wilson 2015 Paleobiology). I think these types of analyses would be more powerful for inferring diets in fossils.

However, an issue with 'projecting' fossils onto a PCA (or running an LDA/CVA) is that I doubt you can easily build on that analysis to produce a morphofunctional landscape (like you did in Figures 3 and 4) or run Procrustes ANOVAs using fossils. So I understand if you decide to keep the current PCA as your primary morphospace analysis. But you should consider running additional, supplemental multivariate analyses, because they may provide more powerful dietary inferences for the fossils.

Our reply: We agree with the reviewer on this point. We have updated the PCA and 'projected' the fossils onto the morphospace of extant taxa. The whole section "Jaw shape and diet in small mammals" has been updated (it is not entirely in blue font for legibility reasons).

We see a very good correspondence between the proposed dietary categories for Mesozoic taxa and their position in the morphospace, including most stem mammals. We were still able to generate the morphofunctional landscape, using the PC scores of the extant taxa and the "projected" scores of the Mesozoic taxa.

After projecting the extinct taxa data onto the extant taxa morphospace, we no longer see the distinction in jaw shape between extinct and extant mammals in the PCA. Therefore, we removed Table 1 (Procrustes ANOVA of jaw shape between extinct and extant taxa) because we did not think it was necessary given the new focus of the paper. We only kept the Procrustes ANOVAs of jaw shape~diet in extant mammals which we still discuss in the paper.

We have also run a phylogenetic flexible discriminant analysis (phylo FDA; see page 12: lines 214-232) based on the recommendation of the reviewer. For this analysis we wanted to see if by using jaw shape and mechanical advantage we could get a separation between dietary categories among extant mammals. Ideally we would have used the PC scores of the jaw shape analysis, considering the principal components that accounted for at least 90% of the variance. However, our groups of extant mammals were too small for the analysis to be statistically sound: our smallest group had 8 taxa (i.e., herbivores), but we needed 12 principal components to explain 90% of the variance (plus the mechanical advantage values). Instead, we used the regression values from the PGLS of jaw shape on mechanical advantage, since we could see a clear separation between dietary groups [Figs. 5 and 6 (note updated figure numbering)]. Related to one of the comments below, we performed this regression using mechanical advantage values where the biting point was the m1, but also considered those where the biting point was the tip of the jaw. We explain why below. By using these data, we found a very good separation between dietary groups among extant taxa, and also a very good correspondence between the previously proposed diets of Mesozoic mammals from the literature and those suggested by the analysis.

Outlever measurement and MAT/MAM results

I'm surprised that you measure the outlever length (moment arm of resistance) to the anterior tip of the jaw, which models a bite at the incisors. I think m1 (and/or the canine) is a more relevant bite point for measuring the outlever, because stronger bites will occur at this position. The m1 location may be a bit uncertain for some Mesozoic taxa, but I think it's still possible to at least have a justified estimate of this position for all taxa.

Related to this issue is that there's overlap in the GMM data and the mechanical advantage data. That is, Landmarks 1, 3, and 5 from the GMM are also used for measurements of the inlevers and outlever. So the mechanical advantage information is already somewhat encompassed in the GMM shape data. And your PGLS regressions of jaw shape vs MAM/MAT don't tell us much, because you're just regressing similar data against each other. If you instead measure the outlever as the distance from the jaw joint to m1 (which isn't a GMM landmark), it helps differentiate the mechanical advantage data from the GMM data. (See comments below.)

I recommend re-calculating mechanical advantages using an outlever that's measured to m1 rather than the anterior tip of the jaw. If you don't follow this suggestion, then you should at least justify your choice of outlever measurement with discussion and citations in the text.

In Grossnickle (2020), I found that diet is more strongly correlated with the posterior length of the jaw (m1 to jaw joint) than the entire jaw length. So an added benefit of measuring the outlever to m1 is that it may improve the power of MAT and MAM to predict diets.

Our reply: We re-calculated all the mechanical advantage measurements using the m1 as the outlever and we found that it didn't discriminate between dietary groups as well as the jaw tip measurement (see Figs. 5 and 6 and m1 PGLS). Therefore we decided to keep the jaw tip mechanical advantage measurements to use in the PGLS in the main text. Having said that, by combining the PGLS regression scores of jaw shape on mechanical advantage, using both the m1 and the jaw tip as outlevers, we got an excellent separation of dietary categories when performing the phylo FDA (see page 11: lines 210-227). Therefore, we have decided to use the mechanical advantage measurements of both m1 and jaw tip as outlevers in the PGLS regressions, and thus we have 4 regressions now (masseter jaw tip, temporalis jaw tip, masseter m1, temporalis m1). **Any mention of the m1 measurements in the manuscript text is in red type.**

We also clarify the situation for some of the landmarks being used for the mechanical advantage measurements in the text (page 7-8, lines 183-186).

MINOR CORRECTIONS

We have made all the corrections indicated in this section, and have made comments throughout where points needed further clarification

Page 1:

Please add line numbers. It makes it easier for reviewers to refer to specific text. **Done.**

Gil should be Gill. **Corrected.**

"dietary performance" seems odd to me. Maybe edit it to "diet", "dietary preference" or "dietary ecology". Or, if you want to include something about function, then maybe "Jaw morphology is closely linked to both diet and biomechanical performance, ...". **Corrected.**

I don't think you need to include the abbreviations "(MAM)" and "(MAT)" in the Abstract. **Removed.**

In the literature, "docodontans" seems to be preferred over "docodonts". This change could be made throughout the paper. I also recommend replacing "eutricodonts" (page 5) with "eutricodontans". **Corrected.**

Page 2:

Some Mesozoic mammals were >5 kg, including *Vintana* (Krause et al. 2014), *Repenomamus* (Hu et al. 2005), and *Didelphodon* (Wilson et al. 2016). Hu et al. (2005) estimate *R. giganticus* to have been 12–14 kg. I haven't seen a mass estimate for *Schowalteria* (Fox & Naylor 2003), but it might have also been >5 kg. Not only is there greater dietary and locomotor diversity in Mesozoic mammals than previously realized (as you note), but there's also greater body size diversity. At minimum, you should replace "all remained small (< 5 kg)" with "most remained small (< 5 kg)", or replace "small (< 5 kg)" with "small-to-medium-sized (< 15 kg)". And you may want to update your citations.

Our reply: The reviewer is right on this point. We have included this sentence in the introduction, lines 14-15:

“In a similar fashion, it was thought that they were restricted to small sizes (< 5 kg), but some taxa, like *Repenomamus giganticus* (approx. 12-14 kg) indicate greater body size diversity among Mesozoic mammals”

Related to this, I notice that you excluded extant taxa that are over 5 kg (page 12). (Maybe you're following Chen & Wilson 2015 and Chen et al. 2019?) This either needs more justification, or I recommend adding extant taxa that are a little larger than 5 kg.

I also noticed that *Repenomamus* is not in the sample for extinct taxa. Both *R. robustus* (Wang et al. 2001) and *R. giganticus* (Hu et al. 2005) preserve nearly complete jaws. And there's more direct evidence for their diets compared to most fossil taxa because of fossilized gut contents that suggest carnivory (Hu et al. 2005). They offer a good test of your analyses' ability to predict fossil diets. I recommend adding *R. robustus* and/or *R. giganticus* to the sample.

We have also included new taxa to our analysis, including five modern taxa over 5 kg, as well as *Repenomamus* and *Gobiconodon*. Additionally, we included a few more taxa with complete jaws that we had missed before or were published recently (i.e., *Argentoconodon*, *Origolestes*, and *Microdocodon*).

Another interesting example of fossil jaw-diet studies is Brannick & Wilson (2018 online, 2020 in print; *J Mamm Evol*), who used the cross-sectional shape of the jaw body to test for durophagy in early metatherians. It's not critical to cite here, but could be added if there's space.

We would have really liked to include this paper; unfortunately, we did not have any more space to do so.

Page 3:

“Comparative mechanical advantage of the jaw has been used ... to study the yaw and pitch of the jaws of Mesozoic therian mammals and relatives [18]”. Technically, in Grossnickle (2017, ref. 18) I used torque instead of mechanical advantage. (I excluded out-lever measurements but incorporated estimated muscle contributions.) This is a minor detail, so it's fine to still cite Grossnickle (2017) here. But you may want to edit the sentence to something like “Comparative mechanical advantage (or a similar biomechanical metric) ...”. Or “Comparative biomechanical metrics such as mechanical advantage ...”.

Our reply: We changed the sentence in the following way (page 2, lines 44-45):

“Comparative mechanical advantage of the jaw (or a similar biomechanical metric) has been used as a proxy for prey choice and feeding ecology in stem mammals, to study the yaw and pitch of the jaws of Mesozoic therian mammals and relatives...”

Page 4:

Carnivores and insectivores differentiate more than I expected. This is an interesting result.

Page 5:

“... the piscivorous *Castorocauda* ...” I reiterate that we aren’t certain of fossil diets – piscivory is just the diet that was hypothesized by Ji et al. (2006). Be careful that you don’t present this as if it’s fact. The dietary inference from your study (insectivory?) might be more accurate.

We have modified the text his way (page 3, lines 81-84):

“According to Ji et al., (2006) the swimming docodontan, *Castorocauda*, has dental features indicative of feeding on aquatic invertebrates and small vertebrates, like fish. *Castorocauda* is often depicted as being carnivorous and, particularly, piscivorous (Ji et al., 2006; Meng et al., 2015; Chen et al., 2019). The jaw shape of *Castorocauda* is similar to that of modern day insectivores, supporting the notion of this docodontan feeding on aquatic invertebrates.”

Page 6:

“... our results show that generally, shorter jaws (negative PC1) have higher mechanical advantage and longer jaws have lower mechanical advantage ...”. I wonder if you’ll still get this result if you measure the outlever to m1 like I suggest above.

Our reply: In general, yes. This is especially true for the temporalis, in Supplementary Fig. S10b most of the taxa on the right hand side of the plot (positive PC1 scores) have dark blue or light blue colours (i.e., medium-low to low mechanical advantage values), while taxa on the left hand side of the plot (i.e., negative PC1 scores) have green to yellow colours (i.e., medium-high to high mechanical advantage values). In the case of the masseter, the top left quadrant of the plot (Supplementary Figure S10a) has the taxa with the highest mechanical advantage values, although some other taxa with high MAM are scattered throughout the plot.

Page 7:

“There is a significant correlation between jaw shape and mechanical advantage of the masseter and temporalis ...”. As I note above, your variables (mechanical advantages and GMM jaw shape data) are not independent because you’re using some of the same landmarks for both analyses. To me, these results are basically saying “jaw shape correlates with jaw shape.” It doesn’t tell us much.

You also say there’s “a tighter predictive relationship between jaw shape and MAT than jaw shape and MAM.” But MAT uses three landmarks from the GMM (1, 3 and 5), and MAM only uses two (1 and 5). So the stronger correlation with MAT might just be an artifact of your measurement choices.

Two suggestions. First, I reiterate that I recommend you re-measure the outlevers to m1 instead of the anterior jaw tip (see comments above). This would make the mechanical advantage data a bit more independent from the GMM data. Second, I would be upfront with readers about this issue. You should acknowledge the similarities in the data, and emphasize that the stronger correlation results for MAT might be an artifact of the methods. You could downplay the correlation results (throughout the paper), because they don’t seem

very meaningful. To me, the more interesting use of the MAMs and MATs is the creation of the morphofunctional landscape (Fig. 4).

Our reply: We re-measured the outlevers to the m1. Regardless of whether the outlever was measured at the m1 or the jaw tip there was a significant correlation between jaw shape and mechanical advantage ($p=0.001$ in all cases). In both scenarios R^2 values were higher in the jaw shape–MAT regression than in the jaw shape–MAM regression. R^2 values are lower when the outlever is measured at the m1.

We included the following sentence in the text (page 7-8, lines 183-186):

“ R^2 values are greater in the plot of jaw shape against mechanical advantage of the temporalis, implying a tighter predictive relationship between jaw shape and MAT than jaw shape and MAM: this might be a result of measuring the temporalis moment arm between two fixed landmarks (Supplementary Fig. S6).

I don't know what the journal's formatting policy is on this, but I recommend writing “Figure” instead of “Fig.” if the figure is discussed in the sentence (rather than cited in parentheses). For example, “In Fig. 7, ...” should probably be “In Figure 7, ...”. This change could be applied throughout the paper.

We have changed Fig. to Figure when discussed in the sentence.

Page 12:

You should provide more justification for excluding allotherians (and rodents). You say it's because of their herbivorous diet and derived jaws, but some allotherians were probably insect- dominated omnivores (e.g. Wilson et al. 2012) with similar diets to many other Mesozoic mammals. You could help justify your decision by pointing out that many allotherians have relatively unique palinal jaw movements (e.g. Von Koenigswald et al. 2012 Pal Z), resulting in unique biomechanics, muscle configurations, and jaw shapes.

Related to this, I recommend adding *Haramiyavia* (Luo et al. 2015 PNAS) to analyses. It doesn't have the strong palinal movement like many other allotherians, and I personally question whether it's even related to later haramiyids from the Middle-Late Jurassic (e.g., see King & Beck 2019/2020 preprint in bioRxiv). My guess is that it'll plot along PC1 near *Morganucodon* and extant insectivores. You don't have to follow this suggestion, but it's worth considering. It'd give you another interesting data point.

Our reply: We have included further justification for excluding allotherians from the sample. The inclusion of allotherians in the sample was also raised by Reviewer 3. We decided to include some allotherians in the analyses to see where they would plot in relation to our sample. Perhaps unsurprisingly (and in line with the results of Grossnickle and Polly, 2013) allotherians plot in their own area of morphospace, far away from other non-allotherian Mesozoic mammals (Supplementary Fig. S7). They were also dissimilar to any extant mammals in our sample. When including them in the PGLS regressions, because of their comparatively higher mechanical advantage, they skewed the rest of the sample and we stopped seeing clear differences between dietary groups (Supplementary Fig. S11).

Having said that, we decided to include *Haramiyavia* in our sample because, as Reviewer 1 points out, it might not be closely related to the Jurassic euharamiyids according to King and Beck, 2020.

The text is modified as follows (page 14-15, lines 318-328):

“For this study, we chose to exclude the Mesozoic haramiyids and multituberculates, and only focus on non-allotherian Mesozoic taxa. Grossnickle and Polly had previously determined that the jaw shape of multituberculates is different from other non-allotherian Mesozoic mammals. We attempted to include allotherians of different diets in the sample, but they all plotted in their own area of morphospace, far away from other non-allotherian Mesozoic mammals. They were also dissimilar to any extant mammals in our sample (Supplementary Figure S7). Compared to non-allotherian Mesozoic mammals and extant mammals of different diets, multituberculates and haramiyids had higher mechanical advantage values, which skewed posterior analyses and prevented us from seeing clear differences between dietary groups among extant mammals (Supplementary Figure S11). Allotherians have a very derived jaw morphology, dissimilar to other Mesozoic mammals and extant small mammals, they also have relatively unique palinal jaw movements (Krause, 1982), which could lead to a unique biomechanical biting performance. Therefore, we have decided to exclude allotherians from this sample, with the exception of *Haramiyavia*, which is probably not closely related to later Jurassic euharamiyids, following King and Beck (2020).”

Page 13:

You cite Close et al. (2015) for the overall tree topology, which is fine. But I think Close et al. co-opted character matrices from other studies for their analyses, so you might consider instead citing the original papers. Or you could cite a more recent phylogeny, such as Huttenlocker et al. (2018, Nature), which continues to show the same overall topology using a bigger sample of fossils.

Our reply: We agree with this and have updated the phylogeny as per Huttenlocker et al., (2018) and also considered King and Beck (2020) regarding the position of *Haramiyavia*. We have updated the methods accordingly

Page 14:

“in in” Corrected.

References:

Ref 18 should include the article number: 45094

Article number has now been included (Note that this is now reference 20)

Ref 22 should include Wible as a co-author.

Corrected (Note that this is now reference 26)

Table/figure captions:

Many captions, especially in the Supplement, are too brief and often lack complete sentences. For example, “Supplementary Fig. 4. Placentalia phylogeny”. You should add more detail.

The caption of these figures has been modified to include more detail. For example, the caption of Supplementary Fig. 4 now says:

“Supplementary Fig. S4. Species level phylogeny of the placental taxa used in this study. Phylogeny assembled from Song et al., 2012 (overall topology), Roberts et al., 2011 (Scandentia), Nyakatura and Bininda-Emonds, 2012 (Carnivora), Amador et al., 2018 (Chiroptera), Springer et al., 2018 (Eulipotyphla), Seiffert, 2007 (Afrosoricida).”

In the captions for Figures 6 and 7, you note “Colours indicate known dietary categories of extant mammals and suggested dietary categories for Mesozoic mammals (obtained from the literature).” But Figure 3 is the first figure to plot taxa with those colours. So I recommend adding this sentence to the Figure 3 caption.

This sentence has been added to the PCA plot figure (now Figure 2).

Text in figures:

Some text is very small and hard to read (e.g. “horizontal line” in Figure 2B). I recommend increasing the font size where possible.

We have increased the font size wherever possible in most figures. Most figures have also been re-designed to increase legibility.

Figure 1:

Consider adding representative jaw silhouettes for each of the major clades in the phylogeny (except placentals and marsupials). I think it’d improve the figure.

We have added jaw silhouettes to Fig. 1

Supplementary Table 1:

Chen et al. (2019) might have used OPC data to help categorize the diets of modern taxa, but I don’t think they used OPC for the diets of fossil taxa. From the Chen et al. 2019 Supplementary Information (pg 4): “To infer the dietary preference of each fossil species in our study we relied on our own assessment of gross dental morphology, wear facets, and gut contents as well as assessments by previous studies (SI Appendix, Table S4).” This should be corrected in the table.

We have corrected this statement. SI Table S4 has been moved to a spreadsheet (i.e., Supplementary Data 1) with additional data on every taxon.

Page after Supplementary Figure 6:

“Si Figure 6” should be Supplementary Figure 6.

Corrected.

Supplementary Figures 8 and 9:

I don't see labels for A and B in the figures.

Corrected.

Dave Grossnickle dmgrossn@uw.edu

REVIEWER 2

The article entitled "Jaw shape and mechanical advantage indicate diet in Mesozoic crown, but not stem, mammals" by Morales-Garcia and co-authors tests whether jaw shape and jaw mechanical advantage are correlated and whether they are correlated to diet in small-bodied extant mammals and in turn in Mesozoic mammals. Ecomorphological studies of Mesozoic mammals have increased in recent years, focusing on various aspects of morphology and ecology (teeth and diet, jaws and diet, postcrania and locomotion, etc.). This study mainly builds on work by Grossnickle and Grossnickle and Polly. It largely uses the same landmarks and extant sample, but adds some methodological twists and some taxa both extant and extinct. It also compares estimates of mechanical advantage of the adductor muscles to shape as measured from GM. Overall, I thought that the paper was well written, the analytical approaches were sophisticated (but sometimes not clearly detailed), the direction of the paper has good potential, but I thought that it lacked depth of explanation and a major evolutionary punchline beyond the methodological implications. I don't know the journal but see that it's in the Nature family of journals. As it presently stands, I would not recommend it for publication in this journal, perhaps after major revision. My detailed comments are below:

Our reply: We appreciate all the comments raised by the reviewer that have helped us strengthen the paper. We address all of their points below.

1) The abstract should have more information in it? How many taxa sampled, which ones? Strikes me as rather vague, light on details.

Our reply: We have added some details on the taxa sampled, but unfortunately the word limit on the abstract (150 words) does not allow us to be more specific. Here is the sentence we added to the abstract:

"Here, we analyse the relationship between form and function in **the jaws of 70 living and 45 extinct mammals spanning the Late Triassic-Late Cretaceous**, using geometric morphometrics and mechanical advantage of the masseter and temporalis."

2) The use of images from the literature should be cautiously approached. There was no discussion of how images were vetted. Images taken of jaws in lateral view can be at slightly different angles and this can affect measurements central to this paper. Was that evaluated? I didn't see mention of it.

Our reply: We have included this sentence in the text (page 15, lines 332-334):

“All photographs were reviewed by NMMG to ensure the jaws were all captured in the same orientation; some photographs were rotated in order to have the horizontal ramus of the jaw parallel to the x axis”

3) The presentation of results can by nature be rather dry. However, rather than series of statements about correlations, I would encourage the authors to try to provide a topic sentence for each paragraph and have that topic sentence give the punchline of what will be detailed in the subsequent sentences. This allows the reader to trudge through the details with some sense of what is meaningful and what should be coming.

Our reply: We have reviewed the text and modified the text in places (see below for examples) to make the findings of each paragraph of results clearer. Because the format of the journal requires that the methods are placed at the end of the article, we do have to add some additional context to each paragraph to make it clear what data were analysed and with what methods.”

Here are some examples of how we modified the text:

- Page 3, lines 56-57: We start the paragraph with the following topic sentence: “Using 2D geometric morphometrics, we found that jaw shape is a good proxy for diet among small extant mammals.”

-Page 3, lines 69-70: We start the description of the results with “Most stem mammals plot within the morphospace of extant insectivores”. We later expand on this, pointing out exceptions and previous hypothesis for the diet of stem mammals. We follow this structure for non-therian crown mammals and therian crown mammals.

-Page 6, lines 138-140: We start the description of the mechanical advantage results with “We obtained mechanical advantage (MA) data to test whether extant mammals of different dietary groups have distinct MA values: while many dietary groups are statistically different from each other (Table 3), mechanical advantage values on their own do not obviously correspond to a particular dietary category.”

4) In the end the results indicate that in extant mammals the GM shape measures and mechanical advantage are significantly correlated to each other and with diet (MAT more than MAM). Many of the results are as you would expect. Notably, GM and mechanical advantage are measuring very similar components (some of the GM landmarks are the same position of MA measurements). So these aspects don't strike me as particularly novel. However, the discrimination of insectivores and carnivores is more novel, so why not spend some more time discussing those results. And that there is no statistical difference b/w herbivore MAT and carnivore MAT. I would encourage the authors to spend more time digging into these aspects of the results.

Our reply: We have added further discussion on these points:

- Regarding discrimination between insectivores and carnivores (page 12, lines 234-243):

“In this study we found that jaw shape is largely indicative of dietary category (with the exclusion of omnivores) in small extant mammals. This reflects the findings of Grossnickle and Polly; however, we find a clearer discrimination of jaw shape between carnivores and insectivores, which might be related to our larger sample size and landmarking regime. In Fig. 2 there is some overlap between these two dietary categories but, in general, hypercarnivores such as the felids (#79,80,81) and the Tasmanian devil (*Sarcophilus harrisi*, #58) are clearly distinguished from the insectivores, while some mesocarnivores like the quolls (genus *Dasyurus*, #56,57) and the Corsac fox (*Vulpes corsac*, #90) are more similar to the insectivores. This is in agreement with the study by Prevosti et al., 2011 who also found a clear separation of hypercarnivorous mammals from mesocarnivores and insectivores. As hypothesized by Prevosti et al and previous authors, hypercarnivores have shorter jaws to increase the mechanical advantage of the adductor musculature and deliver a stronger bite; alongside other morphological features, this configuration proves advantageous for prey subduing and meat consumption.”

- Regarding no statistical difference between herbivore and carnivore MAT (page 12, lines 244-250):

“Generally, it is expected that herbivores would have a high mechanical advantage (i.e., increased bite force) of the masseter (MAM) and that carnivores would have a high mechanical advantage of the temporalis (MAT). Interestingly, we found that herbivores not only tend to have high MAM, but also high MAT, while carnivores have high MAT, but low MAM. A study on cranial morphology of rodents (Samuels, 2009) found a similar pattern: herbivores have enlarged masseter and temporalis muscles, while carnivores have and enlarged temporalis and a reduced masseter. Despite their different diets, the enlarged temporalis muscle in herbivores and carnivores might serve a similar function: to resist dislocating forces when biting at the front of the jaw, either when dealing with struggling prey (in carnivores) or biting hard plant material (in herbivores).”

5) When it comes to the fossil data, why have the authors given more weight to pre-existing dietary inferences. Whose to say that those data are better estimates of the diets of these taxa. This is another area where I thought they should've provided greater discussion. What is the rationale for taking as “truth” the dental inferences or inferences from previous authors?

Our reply: We appreciate this comment and have re-focused the manuscript taking this into consideration. The first reviewer raised a similar issue and we now avoid taking previous inferences as the “truth”, and rather discuss different dietary hypotheses more broadly and project the fossil taxa into morphospace to test pre-existing dietary inferences rather than take these inferences as fact.

6) I couldn't tell from the described methods but it seemed as though they put extant and extinct data together and then ran the PCA. Instead, I would recommend that they run the PCA with only the extant data and then subsequently project the extinct data into that morphospace. In this recommended approach you're essentially predicting diet of the extinct taxa using the jaw shape space formed by extant taxa of known diets. As is, the extinct taxa are strongly influencing the shape space.

Our reply: This comment was also raised by Reviewer 1. We agree that projecting the fossil data onto the morphospace of extant taxa is a much better way of analysing and presenting the data. We have now done this and found a very good correspondence between the proposed dietary categories for Mesozoic taxa and their position in the morphospace. The manuscript has been restructured accordingly.

7) While they mention that stem mammals fall outside of the shape space occupied by extant taxa, in reality it looks as if many extinct taxa (not just stem mammals) fall outside of that shape space. I would've appreciated more discussion of what that means for functional inference. I would argue that the principles of biomechanics still apply. So how can we use these to still constrain dietary inference.

Our reply: Now that we have modified the PCA to 'project' the extinct taxa onto the extant taxa morphospace we do not see this anymore and we have decided to exclude it from our discussion.

8) Ultimately, I felt that the paper could use further digestion of the results and further exploration into what these results mean more broadly for the evolution and paleoecology of Mesozoic mammals.

Our reply: We have modified the conclusions (page 14, lines 296-307) in the following way to include broader implications of our results (see also first two paragraphs of Discussion section):

“These morphological and functional characteristics reflect the differential need of these animals for acquiring and processing particular types of food, which can be aided by having either stronger or faster jaw closure. Other factors, such as different foraging strategies and phylogenetic history, also play a role in determining the morphological configuration and functional traits of the mammalian jaws. The fact that the jaw shape and mechanical advantage of extant mammals could be successfully used as the basis for inferring the diet of mammals living during the Mesozoic (even those with retained postdentaries), highlights that ecological pressures and principles of jaw design were similar today and in the past.

We corroborate the hypothesis that most non-allotherian Mesozoic mammals had a faunivorous diet. Considering the taxa in our sample, most stem mammals, symmetrodonts, dryolestids, amphitheriids and eutherians seem to have had an insectivorous diet (or one consistent of aquatic invertebrates in the case of semi-aquatic mammals such as *Castorocauda* and *Teinolophos*), while the eutriconodontans and metatherians studied here probably had a more carnivorous diet. Fossorial or semi-fossorial mammals with a potentially vermivorous diet can also be distinguished from other insectivores as their jaw shape is more similar to that of carnivores.”

REVIEWER 3

Different morphological traits under selective pressures, like the tooth shape, linear measurements of postcranial elements, were widely used to investigate the

ecomorphospace disparity of extinct groups in the geological history. The authors expanded a morphometric analysis of lower jaw, initiated by Grossnickle and Polly (2013), from linear measurements to geometric morphometrics, and made use of modern analogues, which to assess changes in mammalian morphological and dietary disparity. This geometric morphometrics may increase the applicability of statistical analysis in fossil study and reduce the possible artificial influence. The authors further used the functionally-driven approaches to clarify the relationship between the mechanical advantage and ecomorphospace disparity. The result shows some correlation between the shape and function, and the two traits have the potential to be the proxies to elucidate the dietary preferences of extinct mammals. I would recommend the manuscript for publication if the following points are addressed.

Our reply: We appreciate all the comments by the reviewer to improve our paper, we address their comments below.

1. The study of jaw shape and diet lacks statistical correction for phylogenetic non-independence. The PC1 and PC2, especially PC2 for scatter plots of the Principal Component Analysis, are more or less related to the Phylogeny of the chosen taxa. When we look at the plots of all extinct taxa, most Carnivores, Insectivores, Omnivores are not fallen into the corresponding morphospace of extant taxa. But if we look at those plots fallen into the overlap zone of extant and extinct taxa, most of which belong to the Therian crown mammals, the diet of these taxa show a good congruence with those of extant mammals. Because the dietary categories are made by the data of extant therian mammals, the Therian crown mammals within the overlap zone conform to the PCA, where all taxa belong to therian mammals and no phylogenetic information among them. But obviously, the morphology of jaws is strongly correlated with diet. Like the diet fitness of the Mesozoic mammals are much better when the

Phylogenetic Generalized Least Squares (PGLS) regression was used to test the relationships between jaw shape and mechanical advantage incorporating phylogenetic relationships and phylogenetic signal. Thus, I strongly suggest using a Phylogenetic principal component analysis to perform a PCA considering the phylogenetic non-independence between taxa, which may give us the true information for correlation of the jaw shape variation and diet for extinct mammals, and finally help to know that whether jaw shape could be used as a dietary proxy in Mesozoic mammals.

Our reply: We considered the option of a phylogenetic PCA for the jaw shape data, as suggested by this reviewer, but this came into conflict with the suggestions from Reviewers 1 and 2, who asked us to 'project' the fossil data onto the morphospace of extant taxa. Because we are no longer running a PCA of all taxa combined it proved impossible to use one phylogeny on two separate samples. When using this 'projection' method, we see a very good correspondence between the proposed dietary categories for the Mesozoic taxa and where they are plotting in the morphospace of modern taxa. However, we do use a phylogeny in the PGLS regression and in a new analysis added post-review, which was a Phylogenetic Flexible Discriminant Analysis. This method uses data on jaw shape, mechanical advantage, and phylogenetic relationships to classify the extinct taxa onto one of three dietary categories (i.e., carnivore, insectivore, herbivore) based on the sample of extant taxa.

2. The results of the ANOVA analysis, such as the principal components, should be listed somewhere in the manuscript.

Our reply: The PC scores can be found in the new Supplementary Data 1 spreadsheet. This spreadsheet also includes all the other relevant data used in this paper, including the mechanical advantage values, observed or proposed diets, photograph sources, etc. The results of the statistical analyses can be found in Tables 2 and 3.

3. For the taxa sampling, I feel like if some representatives of multituberculates (especially a few taxa representing herbivores and Omnivores) and haramiyidans included in this exploration, the result would be more informative. For the extinct taxa, there is no herbivory representative in this analysis, and the authors excluded multituberculate because of their derived morphology, while multituberculates, especially the Late Cretaceous multituberculates, is one of the few mammalian groups that could be referred to herbivores in Mesozoic mammals. To be honest, the morphology of lower jaw in multituberculates did not change too much during their evolution. I would suggest use multituberculates to fill the vacancy of lacking herbivores data.

On the other hand, the authors claimed that the anteriorly positioned angle of the mandible is the reason that stem mammals largely plot in a different area of morphospace to extant mammals. About the view of Haramiyidans, or Euharamiyidans from China, their controversy is mainly on the presence of postdentary bones or not. If the analysis contains representatives of two different kinds of the lower jaw from the same groups, a comparison could be used to test the authors' hypothesis of the stem mammals. Since the anteriorly positioned angle of the mandible and retained postdentary bones are the important difference between stem mammals with others, clarify this question is crucial for the usability of this method in stem mammals.

Our reply: This is related to a remark from Reviewer 1; please also see our comments to review 1 in this respect. We decided to include some allotherians in the analyses to see where they would plot in relation to our sample. In line with the results of Grossnickle and Polly, 2013, allotherians plot in their own area of morphospace, far away from other non-allotherian Mesozoic mammals. They were also dissimilar to any extant mammals in our sample. When including them in the PGLS regressions, because of their comparatively higher mechanical advantage, they skewed the rest of the sample and we stopped seeing clear differences between dietary groups (see Supplementary Figures S7 and S11). Therefore, we decided to exclude them from our analyses, but kept these plots in the Supplementary Information to justify their exclusion from the analysis. We decided to include *Haramiyavia* in our main sample because, as Reviewer 1 pointed out, it's likely not closely related to later Jurassic euharamiyids according to King and Beck, 2020.

We modified the text as follows (page 14-15, lines 318-328):

"For this study, we chose to exclude the Mesozoic haramiyids and multituberculates, and only focus on non-allotherian Mesozoic taxa. Grossnickle and Polly had previously determined that the jaw shape of multituberculates is different from other non-allotherian Mesozoic mammals. We attempted to include allotherians of different diets in the sample, but they all plotted in their own area of morphospace, far away from other non-allotherian Mesozoic mammals. They were also dissimilar to any extant mammals in our sample

(Supplementary Fig. S7). Compared to non-allotherian Mesozoic mammals and extant mammals of different diets, multituberculates and haramiyids had higher mechanical advantage values, which skewed posterior analyses and prevented us from seeing clear differences between dietary groups among extant mammals (Supplementary Fig. S11). Allotherians have a very derived jaw morphology, dissimilar to other Mesozoic mammals and extant small mammals, they also have relatively unique palinal jaw movements (Krause, 1982), which could lead to a unique biomechanical biting performance. Therefore, we have decided to exclude allotherians from this sample, with the exception of *Haramiyavia*, which is probably not closely related to later Jurassic euharamiyids, following King and Beck (2020)."

Minor issues:

- In my opinion, the description of Mesozoic mammal plotting in the Supplementary should be an important component of the Discussion. Using the Mesozoic mammal data to solve scientific problems will be of interest to most researchers. Instead, there are some verbose expressions in the Result and Discussion of the main text.

Our reply: We have now moved this description from the Supplementary Information to the main text.

- The authors stated that "Mesozoic mammals are now considered to have been more ecologically diverse, although all remained small (< 5 kg) [1,2]" in the first paragraph (page 1), which is inaccurate. Some taxon, like *Repenomamus* or *Vintana*, could have body mass larger than 5 kg.

Our reply: We agree with this point and have amended the manuscript accordingly. We have also included *Repenomamus* in our sample, and included a few extant taxa over 5kg.

- Typo in the legend of Supplementary Fig. 6: A) Extant taxa; B) Extinct taxa. **Corrected.**

Reviewers' comments:

Reviewer #1 (Remarks to the Author):

See attached file. 
Reviewer #2 (Remarks to the Author):

The authors have done a commendable job on improving their manuscript in revision. In particular, I think that the Results and the Discussion provide a much better supported story that reads more clearly. I have very few comments and, if addressed, I would consider the manuscript acceptable from my perspective. I think it will make a fine contribution to the growing number of ecomorphological studies of Mesozoic mammals.

Specific comments:

1) In either the intro where there is a discussion of previous functional studies or when the Therian crown mammals are discussed (line 128-129) Wilson et al. 2016 and Brannick and Wilson 2020 should be cited. Those papers present biomechanical analyses of the dentaries of Didelphodon and Eodelphis (and Alphadon) and dietary inferences. They are relevant references to this study (more so than ref. 32).

2) Also in the Therian Crown mammals section, line 126 there's a statement that extinct metatherians are considered insectivorous and carnivorous, but in addition to polydolopimorphians from SA there is also the Late Cretaceous NA taxa Glasbius that was likely frugivorous (Clemens 1966, Wilson 2013).

3) The inference that Teinolophus was semiaquatic is far from well supported (based on size of the mandibular foramen only). I would be more cautious with this interpretation.

4) Line 81-84, in the description of results there is mention that previous workers include aquatic invertebrates among the possible diet of Castorocauda. This is correct, but the aquatic invertebrates referenced are hard-shelled invertebrates like those eaten by seals not insects. If I correctly understand your results, the jaw shape of Castorocauda falls among extant insect eating mammals included in your analysis, not durophagivores.

5) Line 216, delete "of" after "taxa"

6) In the Geometric Morphometrics methods, its the partial warp scores that are submitted to a PCA.

Again I appreciate the effort that the authors have made on the revisions. The manuscript is greatly improved.

Sincerely,
Gregory Wilson Mantilla

Reviewer #3 (Remarks to the Author):

Morales-García and co-authors did a great job incorporating comments and providing detailed responses to reviewers' queries. I appreciate their efforts, especially the phylogenetic correction at the last three analysis in the manuscript. The result improved a lot and seemed like be applicable for inferring the diet of most Mesozoic mammals. Based on this work, I believe that they will contribute to the reconstruction of the diet of fossil mammals and further to indicate the ecology in geological history, which is of scientific significance and great interest. I look forward to seeing this paper published.

Minor suggestions, no need to be responded:

1. The purpose of this paper is to indicate the diet of Mesozoic mammals then the main emphasis

of this work (results, discussion, and conclusion) should be the Mesozoic mammals. The portion of the extant data can be simplified and refined to save more space for the Mesozoic mammals. In the future, I think the follow-up work could be published on other journals in the Nature family, who may need the manuscript to be pruned to 3 pages and the result need to be novel enough but not just corroborate previous work.

2. It's all about morphology. The conclusions of previous work were based on experience, whereas the authors based on data and analysis. The authors should be more confident about their own results.

Replies to reviewers

We would like to thank all three reviewers for the feedback on the paper. Please find our reply to all the comments in blue text below and highlighted throughout the main text. Following the comments from Dr. Grossnickle we have decided to eliminate the PGLS regressions from the paper and instead present the mechanical advantage results as a scatter plot of MAM/MAT (x axis) vs. PC1 scores (y axis), as this is the PCA axis in which dietary categories among small mammals are best discriminated. This scatter plot is just for data visualisation purposes, and we have removed any mention of testing the relationship between shape and function, as we realise our data was not adequate for this purpose. However, this scatter plot still supports the main conclusions of our paper. This new visualisation serves a similar purpose to the morphofunctional landscapes, but presents the results in a clearer way; therefore, we have decided to move the landscapes to the SI and brought back the methods figure to the main text, as it will be of use to readers which are not familiar with obtaining mechanical advantage measurements. We reiterate our gratitude to all three reviewers, as you have helped improve this manuscript with all your kind comments.

Reviewer 1

I appreciate that you put considerable effort into revising and improving the manuscript. I really like the added discriminant analysis (pFDA) and revised PCAs, which strengthen your central argument that jaw shape of Mesozoic mammals is indicative of diet. I think the paper will be of interest to a broad readership, and I support it for publication. However, I continue to have a concern with the use of mechanical advantage (MA) in the regressions (and the new discriminant analyses), which I discuss below. Also, I recommend adding a jaw measurements figure (like Figure S6 or a simplified version, e.g. see the top of Figure 2 in Navalón et al. 2019) to the main text because the measurements are very important information for this study. I have a long list of additional comments below, but most are minor suggestions.

Our reply: We really appreciate all the comments from Dr. Grossnickle and we detail below our replies to all his comments. We understand the issue with the PGLS regressions, therefore we have decided to remove them from the paper and instead present a scatter plot of mechanical advantage values (x axis) vs PC1 scores (y axis), as this axis of the PCA describes mainly the length of the jaw and the dietary categories among extant mammals are best discriminated on this axis. The discriminant analysis has been changed as well and now uses the first 7 PCs of the PCA of Procrustes coordinates of jaw shape. We have also moved back to the main text the figure describing the landmarking regime and the mechanical advantage measurements. We also decided to move the morphofunctional landscape to the Supplementary Information because the results it was meant to convey were better presented with the new scatter plots.

I reiterate a point that I made in my previous review – the MA data is jaw shape data. It is not independent of the geometric morphometric (GMM) shape data. The same shape information that's captured by the MA ratios is already captured by the GMM shape outlines. You follow Navalón et al. 2019 on bird beaks for your analyses. But in contrast to your mammal data, the main landmarks for their MA calculations (e.g. jaw joint) are NOT also landmarks in their GMM analysis (see Fig. 2 of Navalón et al. 2019). You repeatedly mention that a combination of jaw shape and MA data is powerful for inferring mammal diets (e.g. Abstract, Line 36, Line 211), but you can simplify this to just jaw shape – all of your data is jaw shape data.

Because of this issue, I still don't think that the regressions of GMM shape data vs MA shape data are informative, and probably not necessary at all. One benefit of the regression plots in Figures 5 and 6 might be that they show how MAs differentiate among some dietary groups – but this could also be shown with simple box-and-whisker plots of the MAMs/MATs of different dietary groups (including both extant and fossil taxa).

Our reply: As mentioned above, we understand our mechanical advantage data is not appropriate to be regressed against jaw shape data; therefore, we have removed the PGLS regressions from our

paper and instead present a visualisation of mechanical advantage values vs PC1 scores (i.e., jaw length) in the form of a scatter plot. This visualisation supports the conclusions of our analysis in a very similar manner to the PGLS regression plots; however, we have now removed any mention of quantifying the relationship between jaw shape and data, and present it simply as a visualisation of jaw length and mechanical advantage values. We have also changed the abstract accordingly.

In your reply to my earlier review you noted that ideally you'd use PCA scores for the pFDA but can't because of a statistical issue – it'd require more PCs (12) to represent 90% of variance than there is sample size for the smallest diet group (8) – and therefore you instead use the regression scores. I'm confused by this. Maybe I don't fully understand what is meant by "regression scores" here (you could clarify this in the Methods), but certainly if you're simplifying the shape data down to one dimension (the y axis of the Fig 5 and 6 regressions are your only regression scores, correct?), then you're incorporating much less than 90% of the shape variance into the pFDA. If so, then you might as well use PC scores for the pFDA because it's simpler and avoids the problem of regressing shape data (GMM) vs shape data (MA). I recommend that you re-run the pFDA using a smaller number of PCs (7 or 8). I'm guessing that it'll still account for ~75% of the shape variance, which is probably more than you're currently incorporating. I strongly suspect that this pFDA will be as powerful as your current regression based pFDA for differentiating dietary groups, but you can doublecheck by comparing the percentages of correctly classified modern taxa into diet categories by both pFDAs.

Our reply: After eliminating the PGLS regressions from our study, we decided to carry out the discriminant analysis using the first seven PC scores of the jaw shape PCA (which account for 81.39% of the variance), with a lambda value of 0.08. Using this configuration, 89.19% of the extant taxa were correctly classified, this was the highest percentage of correct classification we obtained for the extant taxa.

Minor comments:

- Abstract: “Mechanical advantage values on their own are not very informative of dietary ecology.” This statement seems a little too negative based on the statistical results (Table 3). It looks to me like MAM separates out herbivores from other groups and MAT separates insectivores from other groups (besides omnivores). My recommendation is to note these results in the Abstract rather than state that MA values are not informative.
 - **Our reply:** We have changed the abstract accordingly. It now reads: “Insectivores have low MAM and MAT, carnivores have low MAM and high MAT, and herbivores have high MAM and MAT.”
- Abstract: “The combination of both metrics provides a clear separation between insectivores, carnivores and herbivores.” But doesn’t the GMM shape data alone provide a clear separation between these dietary groups (Table 2)? Why use a combination of metrics if one does the job?
 - **Our reply:** We have eliminated this sentence and restructured the abstract. This section now reads: “In extant mammals, jaw shape discriminates well between dietary groups: insectivores have long jaws, carnivores intermediate to short jaws, and herbivores have short jaws. Insectivores have low MAM and MAT, carnivores have low MAM and high MAT, and herbivores have high MAM and MAT. These traits are also informative of diet among Mesozoic mammals (based on previous independent determinations of diet) and set the basis for future ecomorphological studies.”
- Abstract: “This relationship is informative of diet among Mesozoic mammals.” Because this statement is the crux of the paper, I recommend providing some supportive evidence, such as a brief summary of the information in Lines 63-67.
 - **Our reply:** This relates to the previous point by the reviewer, we have changed the abstract as much as we can while trying to remain within the word limit. As above, it now reads: “In extant mammals, jaw shape discriminates well between dietary

groups: insectivores have long jaws, carnivores intermediate to short jaws, and herbivores have short jaws. Insectivores have low MAM and MAT, carnivores have low MAM and high MAT, and herbivores have high MAM and MAT. These traits are also informative of diet among Mesozoic mammals (based on previous independent determinations of diet) and set the basis for future ecomorphological studies.”

- Line 126-127: Maybe not in your sample, but there are extinct metatherians that were likely omnivorous/herbivorous, such as *Glasbius* (Wilson 2013 Paleobiology) and polydolopimorphians (Goin et al. 2016 Brief History of S. American Metatherians). You should revise this statement.
 - **Our reply:** We have incorporated these references into the text (lines 131-132), it now reads: “Extant marsupials have a large diversity of diets, including herbivory, but the extinct metatherians in our sample are considered to have been limited in diet to insectivory and carnivory (note that there are some putatively herbivorous/omnivorous extinct metatherians, like *Glasbius* and polydolopimorphians (Wilson, 2013; Goin et al., 2016)”
- Line 144: Is your MAM in-lever based on the superficial masseter or deep masseter, or does it roughly estimate the MA of both masseters? It may also be worth noting here (or somewhere in the paper) that the MAM is probably very similar to the MA of the medial pterygoid, which also inserts on the angular process.
 - **Our reply:** We have included the following in the Methods section (lines 360-362): “...these measurements roughly estimate the moment arms of both superficial and deep heads of the adductor muscles of the jaw. The moment arm of the medial pterygoid muscle, although not considered here, is probably very similar to that of the masseter.”
- Line 160 (and throughout the paper): “Fig.” should probably be “Figure” when it’s not in parentheses. This edit could be made throughout the paper.

- **Our reply:** We have made this change throughout the paper.
- Line 181 and Line 384: I recommend noting that “jaw shape” is represented by Procrustes coordinates.
 - **Our reply:** The sentence in line 181 was removed and the one in line 384 (now line 369) now reads: “In order to test for significant differences in jaw shape (as represented by Procrustes coordinates)...”
- Lines 185-186 and Lines 267-269: I’m glad that the authors acknowledge why there might be stronger correlation with MAT, but it may be hard for readers to understand this point without more details. You might note that the MAM calculation includes a measurement that is not based on one of the GMM landmarks (unlike the MAT calculation). Or, as I suggest elsewhere, the regressions of GMM data vs MA data (and related text like this here) could be removed from the paper entirely, or relegated to the supplement.
 - **Our reply:** As mentioned above, we have decided to eliminate the PGLS regressions from the paper; therefore, the sentences the reviewer highlights have been eliminated.
- Figure 4: To help readers who aren’t familiar with mammal jaw morphology, you could provide a small jaw image with the MAM measurements in the upper left corner, and a small jaw image with the MAT measurements in the upper right corner. Also, it might be helpful to indicate on the jaw images where the temporalis and masseter muscles are (e.g. see the force vector arrows in figures in Grossnickle 2017 Sci Rep).
 - **Our reply:** We have moved back the figure describing the landmarking regime and the mechanical advantage measurements to the main text.
- Lines 218-220: Were these taxa misclassified by the pFDA? More broadly, what percentage of modern taxa were correctly classified by the pFDA? This information should be included because it indicates how powerful the analysis is for inferring diets from morphology. (I think

Lars Schmitz' tutorial on his github page has code for this.) You could also consider recording the extant taxa classifications in your supplementary data file (like you did for extinct taxa).

- **Our reply:** We have included the following sentences (lines 150-156): “We used the extant taxa of known diets as the training dataset for the discriminant analysis: these taxa were classified correctly 89.19% of the time. For the most part, we see a good separation between dietary groups among extant mammals (Figure 3), with some exceptions: the primarily herbivorous olingo (*Bassaricyon gabbii*, #94) plots within the carnivores ... and a couple of insectivores plot very near the carnivores (i.e., the little brown bat (*Myotis lucifugus*, #104) and the Hispaniolan solenodon (*Solenodon paradoxus*, #109). These three taxa, alongside the carnivorous greater bulldog bat (*Noctilio leporinus*, #101), were the only extant taxa misclassified by the discriminant analysis”. We have also included the extant taxa classifications in our supplementary data file.

Line 222: I recommend replacing “Figure 7b corroborates” with something like “the fossil results of the phylo FDA (Fig. 7b) corroborate”.

- **Our reply:** We have modified the text accordingly (lines 157-158), it now reads: “The Mesozoic mammals included in our sample have largely been considered faunivorous and the results of the phylo FDA (Figure 7b) corroborate this hypothesis.”
- Lines 244-253: At this point I want to clarify that although I disagree with using the MAs for the regressions and pFDAs, I do think that the MA data are interesting and worth reporting. I like the discussion in this paragraph about the different results for MAM and MAT.
 - **Our reply:** We appreciate the reviewer's comment, and think that the new scatter plots of MA vs PC1 scores (i.e., jaw length), do a good job of conveying the point we wanted to get across without performing a regression that, as the reviewer points out, would have been invalid.

- Line 244-245: Why is this expected? I recommend adding a little more information here, or at least cite some relevant references.
 - **Our reply:** We included the following (lines 247-249): “Generally, it is expected that herbivores would have a high mechanical advantage (i.e., increased bite force) of the masseter (MAM), as they need strong bites at low gapes, and that carnivores would have a high mechanical advantage of the temporalis (MAT), as they need strong bites at wide gapes (see Santana et al., 2010 and references therein).”
- Line 246: The low MAM of carnivores might be related to the superficial masseter needing to attach close to the jaw joint to allow a wide gape, thus reducing the in-lever length (Herring & Herring 1974 Am Nat, and a brief discussion on this Grossnickle 2020 Evolution).
 - **Our reply:** We have updated the text (lines 257-260) as follows “Grossnickle (2020) posited that faunivorous taxa benefit from having a shorter masseter (which attaches closer to the jaw joint than in herbivores) as it reduces the length of the in-lever (consequently lowering the mechanical advantage of the masseter) and allows them to have a wider gape”
- Line 254: “Mechanical advantage values are closely related to jaw shape”. The MA measurements are dimensions of jaw shape, so to me this is like saying “Aspects of jaw shape are closely related to jaw shape.” I recommend removing or revising statements like this.
 - **Our reply:** We have removed this statement and shortened the discussion to exclude any mention of the “relationship” of jaw shape and mechanical advantage.
- Line 264: You probably don’t need to include the “JAPr” abbreviation because you don’t use it later in the paper.
 - **Our reply:** We have deleted this abbreviation

- Line 277-279: It could also be worth noting that the angular process of stem mammals may not be homologous to the angular process of therians (e.g. see brief note on this with references in Grossnickle 2017, pg 4).
 - **Our reply:** This has been noted in the text (lines 266-267): “Stem mammals have a very anteriorly positioned angle of the mandible (not that the angular process of stem mammals might not be homologous to that of therians [see Grossnickle, 2017 and references therein])”

- Line 301: I would avoid the word “design”. Maybe replace with “evolution” or “biomechanics”.
 - **Our reply:** We have modified this in the text (lines 287-289), it now reads: “...highlights that ecological pressures and jaw biomechanical principles were similar today and in the past.”

- Line 302: You probably shouldn’t wait until the Conclusions to first mention that your sample is “non-allotherians”. I realize that this in part because the Methods section that explains your sample is after the Conclusions, but I would still make a note about allotherians earlier in the paper. And in the Results section you could mention the supplemental results with allotherians.
 - **Our reply:** We have included additional mentions of this in the Introduction and Results
 - **Introduction (lines 49-52):** “While the jaw shapes of many multituberculates indicate a herbivorous or omnivorous diet (Grossnickle and Polly, 2013), there is no clear consensus on the diets of many Mesozoic taxa typically considered as ‘generalized insectivores’. Here we include only such generalized taxa and exclude multituberculates and haramiyidans (i.e., allotherians).”

- **Results (lines 68-69):** “See Supplementary Figure S6 for a principal components analysis scatter plot which includes multituberculates and haramiyidans; these taxa were excluded from our study because the vast majority of them have jaw shapes dissimilar to the other extinct and extant mammals in our sample (i.e., allotherians have shorter jaws and thus more negative PC1 scores).”
- Line 318: I believe that “haramiyidans” is preferred over “haramiyids”.
 - **Our reply:** We have made this change throughout the text.
- Line 328: I would replace “is probably not” with “may not be” – there’s still considerable debate about the phylogenetic positions of haramiyidans. And more important than phylogenetic position is probably the biomechanics; early haramiyidans like *Haramiyavia* might have had bites that were more orthal than palinal (like other mammaliaforms), in contrast to later haramiyidans (e.g. eleutherodontids) and multituberculates with more developed palinal occlusion (e.g. Butler & Hooker 2005 Acta Palaeontol Pol). So, if there’s space you could consider making this point as well to help justify your inclusion of *Haramiyavia*.
 - **Our reply:** We have modified the text accordingly (lines 314-316), it now reads:

“Therefore, we decided to exclude allotherians from this sample, with the exception of *Haramiyavia*, which may not be closely related to later Jurassic euharamiyidans, following King and Beck (2020). Additionally, early haramiyidans (like *Haramiyavia*), might have not had the palinal jaw movements of later allotherians (Butler and Hooker, 2005).”
- Line 350: For modern clades, are the branch lengths (and topology) from the cited references? Similarly, is the time-scaling method (Line 353) just used for the fossil branches of the tree?

- **Our reply:** For the modern clades, only the topology was obtained from the cited references (not the branch lengths). The branch lengths of all taxa were obtained using the time-scaling method by indicating the first and last appearance dates. For fossil data, they were largely obtained from the Palaeobiology Database. In the case of the modern taxa both FAD and LAD were set to zero because 1) we could not find FAD of every extant taxon and 2) because if we did not set their FAD to zero the end of their branches did not finish at 0 myr. Additionally, we constrained most of the nodes by using information of the time-scaled phylogenies from the references cited (for both extinct and extant taxa).
- Lines 371-374: I realize that it's common for studies to use these measurements to estimate MAM and MAT. But it may be worth briefly noting some potential limitations of these measurements. For example, the measurements are specifically the moment arms for pitch rotation (i.e. an orthal bite); the moment arms for yaw or roll are in different 3D positions (Grossnickle 2017). Because some mammals have chewing motions that are dominated by yaw (e.g. cows) or roll (e.g. tenrecs), pitch moment arm lengths may not be very relevant to some taxa. Also, you're assuming that the pitch axis of rotation is at the jaw joint (i.e. you measure the in-lever and out-lever to the jaw joint), but the position of this axis is probably not at the jaw joint in many taxa (e.g. see Figure 3 in Iriarte-Diaz et al. 2017 Zoology on primates). These issues are more reasons why I recommend de-emphasizing MA results in this study.
 - **Our reply:** We added the following to the Methods section (lines 364-366): "A limitation of this technique lies on its 2D approach: using this method, we can only calculate the moment arms for pitch rotation, while any three-dimensional movement of the jaw (i.e., yaw or roll) cannot be quantified (see Grossnickle, 2017). Additionally, these measurements assume that the position of the pitch axis of rotation is at the jaw joint in all taxa."

- Lines 387-393: What was the lambda (or 'optimal lambda') value used in the pFDA? It's worth noting this value because the chosen lambda value can significantly influence results (e.g. see Figure 3 in Angielczyk & Schmitz 2014 Proc B).
 - **Our reply:** We have included the following in the Methods section (lines 378-380):
 “This analysis was performed by using the first 7 PC scores of the PCA of Procrustes coordinates of jaw shape, which together account for 81.39% of the variance; a lambda value of 0.08 was used. By using this configuration, 89.19% of the extant taxa were classified correctly.”
- Line 391: “source data”? Maybe you mean source code or scripts?
 - **Our reply:** We have modified the text (line 377), it now says “source code”

Reviewer 2

The authors have done a commendable job on improving their manuscript in revision. In particular, I think that the Results and the Discussion provide a much better supported story that reads more clearly. I have very few comments and, if addressed, I would consider the manuscript acceptable from my perspective. I think it will make a fine contribution to the growing number of ecomorphological studies of Mesozoic mammals.

We really appreciate that Prof. Wilson supports this paper for publication and all his comments.

Please find below our detailed replies:

Specific comments:

- In either the intro where there is a discussion of previous functional studies or when the Therian crown mammals are discussed (line 128-129) Wilson et al. 2016 and Brannick and Wilson 2020 should be cited. Those papers present biomechanical analyses of the dentaries of Didelphodon and Eodelphis (and Alphadon) and dietary inferences. They are relevant references to this study (more so than ref. 32).
 - **Our reply:** We have included these references in the text. It now reads as follows:

- **Introduction (lines 28-30):** “With respect to Mesozoic mammals, ...functionally-informed studies include analyses of jaw ratios (Benevento et al., 2019), skull and jaw mechanics and tooth wear (Gill et al., 2014; Wilson et al., 2016; Brannick and Wilson, 2020)”.
 - **Results (lines 135-138):** “Dental microwear indicates a broad diet consisting of vertebrates, plants, and hard-shelled invertebrates for *Didelphodon*; biomechanical analyses of its skull and jaw points towards a durophagous diet (Wilson 2016, Brannick and Wilson, 2020). Biomechanical analyses of the resistance to torsion and bending of *Eodelphis* jaws, points to a durophagous diet in *Eodelphis cutleri* and non-durophagous faunivory for *Eodelphis browni*. (Brannick and Wilson, 2020)”
- Also in the Therian Crown mammals section, line 126 there's a statement that extinct metatherians are considered insectivorous and carnivorous, but in addition to polydolopimorphians from SA there is also the Late Cretaceous NA taxa *Glasbius* that was likely frugivorous (Clemens 1966, Wilson 2013).
 - **Our reply:** We have revised the text to incorporate these taxa (this point was also raised by reviewer 1), the text now reads (lines 131-132): “Extant marsupials have a large diversity of diets, including herbivory, but the extinct metatherians in our sample are considered to have been limited in diet to insectivory and carnivory (note that there are some putatively herbivorous/omnivorous extinct metatherians, like *Glasbius* and polydolopimorphians (Wilson, 2013; Goin et al., 2016)”
- The inference that *Teinolophos* was semiaquatic is far from well supported (based on size of the mandibular foramen only). I would be more cautious with this interpretation.
 - **Our reply:** We have revised the text accordingly, the sentences previously stating *Teinolophos* was semi-aquatic have been modified and now read:

- Lines 107-108: “It has been proposed that the Early Cretaceous monotreme, *Teinolophos*, had a semi-aquatic lifestyle (on the basis of its enlarged mandibular canal, Crumpton [2013])...”
 - Lines 86-87: “The other Mesozoic mammal in our sample proposed to have been semi-aquatic, *Teinolophos*...”
 - Lines 292-293: “... in the case of mammals proposed to have been semi-aquatic, such as *Castorocauda* and *Teinolophos*.”
- Line 81-84, in the description of results there is mention that previous workers include aquatic invertebrates among the possible diet of *Castorocauda*. This is correct, but the aquatic invertebrates referenced are hard-shelled invertebrates like those eaten by seals not insects. If I correctly understand your results, the jaw shape of *Castorocauda* falls among extant insect eating mammals included in your analysis, not durophagivores.
 - **Our reply:** We have clarified these statements in the text, they now read:
 - Lines 85-86: “The jaw shape of *Castorocauda* is similar to that of modern day insectivores; this docodontan might have been feeding on ‘soft’ aquatic invertebrates”
 - Lines 292-293: “...or one consistent of ‘soft’ aquatic invertebrates, in the case of mammals proposed to have been semi-aquatic, such as *Castorocauda*...”
- Line 216, delete "of" after "taxa"
 - **Our reply:** We have deleted it
- In the Geometric Morphometrics methods, its the partial warp scores that are submitted to a PCA.
 - **Our reply:** We used the aligned landmark coordinate data, but we have modified the text accordingly to make it clearer (lines 353-354). It now reads “The resulting

Procrustes aligned landmark coordinate data of the extant taxa were submitted to a principal component analysis (PCA) in RStudio..."

Reviewer 3

Morales-García and co-authors did a great job incorporating comments and providing detailed responses to reviewers' queries. I appreciate their efforts, especially the phylogenetic correction at the last three analysis in the manuscript. The result improved a lot and seemed like be applicable for inferring the diet of most Mesozoic mammals. Based on this work, I believe that they will contribute to the reconstruction of the diet of fossil mammals and further to indicate the ecology in geological history, which is of scientific significance and great interest. I look forward to seeing this paper published.

Our reply: We really appreciate the kind comments of the reviewer and address their suggestions below.

Minor suggestions, no need to be responded:

- The purpose of this paper is to indicate the diet of Mesozoic mammals then the main emphasis of this work (results, discussion, and conclusion) should be the Mesozoic mammals. The portion of the extant data can be simplified and refined to save more space for the Mesozoic mammals. In the future, I think the follow-up work could be published on other journals in the Nature family, who may need the manuscript to be pruned to 3 pages and the result need to be novel enough but not just corroborate previous work.
 - **Our reply:** We have tried to the best of our abilities to simplify the paper and be more concrete in this revised version, we consider the logical structure of the paper to have improved following these two rounds of revisions. We consider essential to discuss how jaw shape and mechanical advantage relate to diet in extant mammals to have strong comparative basis for this study and for future studies, not only of Mesozoic mammals, but also of other small Cenozoic taxa.

- It's all about morphology. The conclusions of previous work were based on experience, whereas the authors based on data and analysis. The authors should be more confident about their own results.
 - **Our reply:** We really appreciate the reviewer's comment and presented our results in a more confident manner.